# TabINR: An Implicit Neural Representation Framework for Tabular Data Imputation

## Abstract

Tabular data builds the basis for a wide range of applications, yet real-world datasets are frequently incomplete due to collection errors, privacy restrictions, or sensor failures. As missing values degrade the performance or hinder the applicability of downstream models, and while simple imputing strategies tend to introduce bias or distort the underlying data distribution, we require imputers that provide high-quality imputations, are robust across dataset sizes and yield fast inference. We therefore introduce TabINR, an auto-decoder based Implicit Neural Representation (INR) framework that models tables as neural functions. Building on recent advances in generalizable INRs, we introduce learnable row and feature embeddings that effectively deal with the discrete structure of tabular data and can be inferred from partial observations, enabling instance adaptive imputations without modifying the trained model. We evaluate our framework across a diverse range of twelve real-world datasets and multiple missingness mechanisms, demonstrating consistently strong imputation accuracy, mostly matching or outperforming classical (KNN, MICE, MissForest) and deep learning based models (GAIN, ReMasker, DiffPuter, CACTI, GRAPE, UnIMP), with the clearest gains on high-dimensional datasets.

## 1 Introduction

Tabular data is a dominant format in healthcare, finance, and the social sciences (Shwartz-Ziv & Armon, 2022), where missing values are ubiquitous and can severely degrade downstream model performance. Poor handling of missingness not only harms predictive accuracy but may also introduce bias with real-world consequences, making robust imputation essential for trustworthy tabular learning and decision making (Rubin, 1976).

Imputation in tabular data is particularly challenging. Tables mix heterogeneous feature types (continuous, categorical, ordinal) and complex nonlinear dependencies (Shwartz-Ziv & Armon, 2022), yet lack the spatial or sequential structure that vision and language models exploit. Many real-world datasets are also small relative to their dimensionality, which complicates generalization for complex models.

A further difficulty is the mechanism of missingness (Rubin, 1976). In `MCAR`, missingness is independent of both observed and unobserved data but is rarely realistic. In `MAR`, it depends only on observed variables and thus requires accurate modeling of conditional dependencies. In the most challenging `MNAR` case, missingness depends on the unobserved values themselves, making unbiased estimation generally impossible without strong assumptions or explicit modeling of the missingness process (see Figure 1). These factors make tabular imputation a central open problem for robust machine learning on structured data.

In this paper, we propose using Implicit Neural Representations (INRs) (Xie et al., 2022) for tabular data imputation. INRs represent data, in our case tables, as neural functions that map coordinates (e.g., row and column indices) to the corresponding values. We believe that such a neural data representation is a natural fit for imputation for several key reasons. First, INRs can inherently fit data even when it is sparse or irregularly sampled. Once trained, the representation can be evaluated across the entire input domain, enabling the imputation of missing entries. Moreover, recent advances in generalizable INRs allow them to capture statistical regularities across datasets while still adapting to individual unseen rows through auto-decoder–style latent optimization (Park et al.,

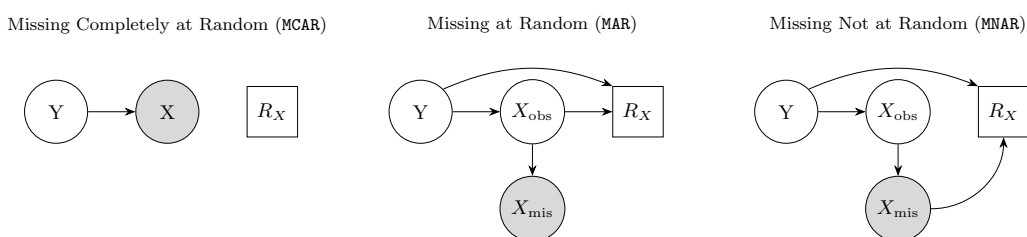

Figure 1: Directed acyclic graphs illustrating common missing-data mechanisms for a partially observed variable $X$. White circles represent fully observed variables, gray circles represent unobserved or partially observed components, and squares denote the missingness indicators $R_X$. Arrows indicate causal or probabilistic dependence.

2019). Unlike many conventional methods, INRs do not rely on strong distributional assumptions or arbitrary discretization choices. Instead, they learn a flexible representation capable of modeling complex dependencies directly from the data. Additionally, INRs are typically built using relatively simple Multilayer Perceptrons (MLPs), which results in a lightweight and fast solution. Although INRs have been widely applied to images (Sitzmann et al., 2020), time series (Fons et al., 2022), and 3D scenes (Mildenhall et al., 2021), their application to tabular data, together with the unique challenges and opportunities this entails, remains largely unexplored. Our mian contributions are:

1. We propose TABINR, a unified INR framework for tabular data imputation that leverages learnable row and feature embeddings to represent instances and variables in a shared latent space.

2. We introduce an auto-decoder-style test time latent optimization procedure that infers personalized embeddings for unseen instances from partial observations, enabling adaptability under sparse conditions.

3. We conduct extensive benchmarks against classical and deep learning baselines, showing that INR-based models achieve competitive or superior performance across diverse imputation tasks while offering a conceptually simple and memory-efficient alternative to GAN and transformer-based approaches.

## 1.1 RELATED WORK

**Data Imputation Strategies**    A variety of approaches have been proposed for imputing missing values in tabular datasets. Early strategies rely on simple heuristics such as mean or mode substitution (Little & Rubin, 2019), expectation–maximization (Jerez et al., 2010), or matrix completion techniques (Hastie et al., 2015). More sophisticated statistical methods include MICE (Van Buuren & Oudshoorn, 1999), MissForest (Stekhoven & Bühlmann, 2012), and MIRACLE (Kyono et al., 2021), which iteratively model conditional dependencies between features. These methods are efficient and widely used, but they either ignore complex nonlinear patterns or depend on correct model specification, while poorly scaling to large datasets (White et al., 2011; Shah et al., 2014).

Deep generative models take a different perspective by modeling the joint distribution of all features. GAN-based approaches such as GAIN (Yoon et al., 2018) and GAMIN (Yoon & Sull, 2020) treat imputation as a conditional generation problem, while VAE-based methods such as MIWAE (Mattei & Frellsen, 2019) and HI-VAE (Nazabal et al., 2020) apply probabilistic inference to heterogeneous tabular data. Although expressive, these methods typically require large datasets and their adversarial or variational training objectives are often unstable and sensitive to hyperparameter choices.

More recently, masked modeling and transformer-based architectures such as TabTransformer (Huang et al., 2020), FT-Transformer (Gorishniy et al., 2021), SAINT (Somepalli et al., 2021), and ReMasker (Du et al., 2024) have demonstrated strong performance on imputation and downstream prediction. In parallel, several powerful modern imputation frameworks have emerged: CACTI (Gorla et al., 2025), a copy-masking transformer architecture; GRAPE (You et al., 2020), which leverages graph neural message passing for missingness-aware representation learning; DiffPuter

(Zhang et al., 2025), which applies diffusion models to tabular imputation; and UnIMP (LLM-Imputer) (Wang et al., 2025), which frames imputation as high-order message passing enhanced by large language models (LLM-Imputer). While these models achieve state-of-the-art performance, they are often heavily overparameterized, incurring substantial computational costs and limiting their scalability, especially on smaller or noisier datasets.

**Implicit Neural Representations**    Implicit Neural Representations (INRs) (Xie et al., 2022; Essakine et al., 2024) model data as continuous neural functions, that map from input coordinates, e.g., row and column indices, to the corresponding value. They have been applied to a large variety of different data modalities like sound (Sitzmann et al., 2020), images (Saragadam et al., 2023), shapes (Park et al., 2019), videos (Chen et al., 2022), or 3D scenes (Mildenhall et al., 2021), and have widely been adopted for a large variety of different tasks. Recent research has primarily focused on addressing the spectral bias (Rahaman et al., 2019) inherent to MLPs typically used in INRs. Solutions include input embeddings such as Fourier Features (Tancik et al., 2020), specialized activation functions like SIREN (Sitzmann et al., 2020), Wire (Saragadam et al., 2023), Gauss (Ramasinghe & Lucey, 2022), HOSC (Serrano et al., 2024), and SINC (Saratchandran et al., 2024), as well as multi-resolution hash-grid encodings like InstantNGP (Müller et al., 2022). Other approaches explore training strategies to identify the most informative samples (Kheradmand et al., 2024; Tack et al., 2023) or techniques for optimal network initialization (Kania et al., 2024). While most INR research focuses on single-instance representations without cohort priors, generalization across multiple instances (Park et al., 2019; Dupont et al., 2022; Bieder et al., 2024; Friedrich et al., 2025) has also been studied. For example, DeepSDF (Park et al., 2019) introduced the auto-decoder framework, which jointly optimizes network weights and signal-specific latent vectors. This framework allows the network to learn patterns shared across different instances, while new signals can be efficiently fitted by optimizing only a new latent vector with the network weights kept frozen.

## 2 METHOD

We propose a unified INR framework, shown in Figure 2, for modeling tabular data. Rather than treating missingness as a nuisance, we parameterize the table as a neural function conditioned on row and feature embeddings. For each cell $(i, j)$, the model maps a learnable row embedding and a learnable feature embedding to a scalar value, enabling a single network to support imputation, downstream prediction, and instance-specific inference.

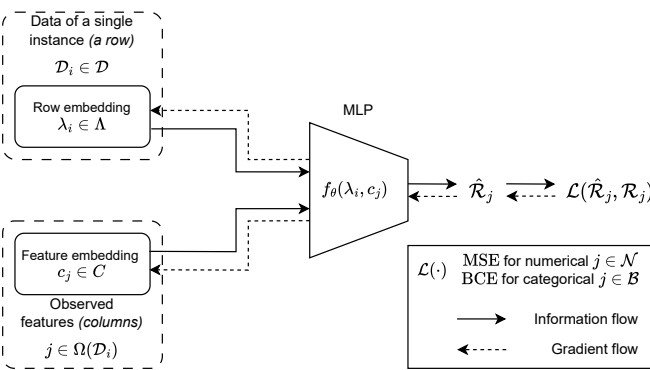

Figure 2: The proposed TABINR framework. During training, we jointly optimize the network $f_\theta$, the row embeddings $\Lambda$, as well as feature embeddings $C$. Once the network is trained, new instances can be added by only optimizing a new row embedding $\lambda_{\text{new}}$, keeping $f_\theta$ and $C$ fixed.

### 2.1 TABINR

Let $\mathcal{D} \in \overline{\mathbb{R}}^{n \times m}$ denote a tabular dataset with $n$ rows (instances) and $m$ columns (features), where $\overline{\mathbb{R}} = \mathbb{R} \cup \{\emptyset\}$ and $\emptyset$ denotes a missing entry. For each row $i$, the observed feature indices are denoted as

$$\Omega(\mathcal{D}_{i:}) = \{j \mid \mathcal{D}_{ij} \neq \emptyset\}. \tag{1}$$

Instead of treating the table as a static array, we model each cell value $\mathcal{D}_{ij}$ as the output of a neural function conditioned on a pair of embeddings:

$$\hat{\mathcal{D}}_{ij} = f_\theta(\lambda_i, c_j), \tag{2}$$

where:

- $\lambda_i \in \Lambda$ is the row embedding representing instance $i$,
- $c_j \in C$ is the feature embedding representing column $j$,
- $f_\theta$ is a shared neural network (e.g. an MLP) that maps the embedding pair $(\lambda_i, c_j)$ to a scalar output $\hat{\mathcal{D}}_{ij}$.

During training, only observed pairs $(i,j) \in \Omega(\mathcal{D}_{i:})$ contribute to the loss. For numerical features, mean squared error (MSE) is used, while for categorical features, expanded via one-hot encoding (OHE), binary cross-entropy (BCE) with logits is applied. This formulation allows the same architecture to handle both feature types within a unified objective.

To identify a suitable network architecture, we conducted a grid search over key hyperparameters. The final configuration, indicated in **bold**, was selected based on the lowest average validation loss across all datasets. Specifically, we varied the row and feature embedding dimensions $\{16, \mathbf{32}, 64, 128, 256\}$, hidden layer widths $\{64, 128, \mathbf{256}, 512, 1024\}$, number of hidden layers $\{\mathbf{2}, 3, 4, 5, 10, 20\}$, dropout rates between $0.0$ and $\mathbf{0.1}$, activation functions $\{\text{ReLU}, \mathbf{SIREN}, \text{WIRE}, \text{SINC}\}$, and learning rates $\{10^{-2}, \mathbf{10^{-3}}, 10^{-4}\}$. A full overview of the hyperparameter search spaces used for all baseline methods is provided in the Appendix (Table 2). We emphasize that this choice reflects a trade-off between stability and generalization across heterogeneous benchmarks, rather than the absolute optimum on any single dataset. Later ablation studies (Appendix Table 8) demonstrate that larger models can achieve marginally better results on individual datasets. The full per-dataset hyperparameter configurations for all baselines are provided as a YAML file in the supplementary material.

## 2.2 TRAINING STRATEGY

We define the set of observed entries as

$$\mathcal{O} = \{(i,j) \mid \mathcal{D}_{ij} \neq \emptyset\}. \tag{3}$$

Training is performed only on this subset, ensuring that missing entries never directly contribute to the objective. For each observed pair $(i,j)$, the model predicts

$$\hat{\mathcal{D}}_{ij} = f_\theta(\lambda_i, c_j), \tag{4}$$

where $\lambda_i$ is the row embedding of instance $i$ and $c_j$ is the feature embedding of feature $j$. This formulation can be seen as a nonlinear matrix factorization: the model learns a shared latent space that jointly captures row-level structure and feature-level dependencies. Because the INR represents all entries through the same neural function, it naturally encodes global cross-feature relationships that are difficult for local methods (KNN, MICE) or iterative tree ensembles (MissForest) to capture especially in high-dimensional tables or under MAR missingness, where conditional dependencies must be modeled rather than approximated locally.

To account for heterogeneous feature types, we use a mixed loss. Let $\mathcal{N}$ be the set of numerical features and let each original categorical feature $g$ be expanded into a one-hot group $\mathcal{B}_g = \{j_1, \ldots, j_{K_g}\}$ of binary columns. Denote by $\mathcal{B} = \bigcup_g \mathcal{B}_g$ the set of all binary one-hot columns. The loss is

$$\mathcal{L}(\theta, \Lambda, C) = \frac{1}{|\mathcal{O}|} \sum_{(i,j) \in \mathcal{O}} \begin{cases} \left(\hat{\mathcal{D}}_{ij} - \mathcal{D}_{ij}\right)^2, & j \in \mathcal{N}, \\ -\left[\mathcal{D}_{ij} \log \sigma(\hat{\mathcal{D}}_{ij}) + (1 - \mathcal{D}_{ij}) \log\left(1 - \sigma(\hat{\mathcal{D}}_{ij})\right)\right], & j \in \mathcal{B}, \end{cases} \tag{5}$$

where $\sigma(\cdot)$ is the logistic sigmoid. Thus, for a categorical feature with $K_g$ categories, we compute one BCE term per one-hot component. Predicted logits are projected back to a valid category via

$\arg\max$ within each one-hot group (winner-takes-all) during inference. Optimization used Adam (Kingma, 2014) with cosine annealing learning rate scheduling and early stopping based on validation loss. During training, we applied random masking of 10-70 % of entries to simulate missingness and evaluate reconstruction fidelity.

## 2.3 TEST TIME ADAPTATION VIA LATENT OPTIMIZATION

At inference, we may encounter a new row $P \in \overline{\mathbb{R}}^m$ with only a subset of features observed, denoted by $\Omega(P)$. Since this row has no pretrained embedding, we introduce a fresh row embedding $\lambda_{\text{new}}$ (initialized randomly) and optimize it while keeping the network parameters $\theta$ and feature embeddings $\{c_j\}$ fixed:

$$\lambda_{\text{new}} = \arg\min_{\lambda} \sum_{j \in \Omega(P) \cap \mathcal{N}} \big(f_\theta(\lambda, c_j) - P_j\big)^2 + \sum_{j \in \Omega(P) \cap \mathcal{B}} \text{BCEWithLogits}\big(f_\theta(\lambda, c_j), P_j\big). \quad (6)$$

At inference, imputing a new row requires optimizing only a single latent code $\lambda_{\text{new}}$ while keeping $f_\theta$ and all feature embeddings $\{c_j\}$ fixed. A few gradient steps on the observed entries adapt $\lambda_{\text{new}}$, after which missing features are imputed via $f_\theta(\lambda_{\text{new}}, c_j)$. No model parameters are updated.

## 3 EXPERIMENTS AND RESULTS

**Datasets & Preprocessing**   We benchmarked TABINR against commonly used imputers including mean/mode imputing, K-Nearest Neighbor (KNN), multiple imputation by chained equations (MICE) (Van Buuren & Oudshoorn, 1999), MissForest (Stekhoven & Bühlmann, 2012), ReMasker (Du et al., 2024), and GAIN (Yoon et al., 2018), as well as several recent state-of-the-art methods such as CACTI (Gorla et al., 2025), GRAPE (You et al., 2020), DiffPuter (Zhang et al., 2025), and LLM-Imputer (Wang et al., 2025). We evaluate our method on twelve publicly available real-world benchmarks spanning different domains, with datasets drawn from the UCI Machine Learning Repository (Dua & Graff, 2017). These datasets vary widely in sample size, dimensionality, and feature composition, covering low- and high-dimensional features as well as mixed numerical and categorical variables. A detailed summary of dataset statistics, including size and number of features, is provided in the Appendix (Table 3). This enables a systematic evaluation of TABINR across heterogeneous tabular data. All baseline models were tuned using the hyperparameter ranges and procedures recommended in their original publications, ensuring a fair and balanced comparison. All datasets are split into 70% training, 10% validation, and 20% test rows. Only training rows are used to learn the model parameters $f_\theta$, the feature embeddings $\{c_j\}$, and their associated row embeddings $\{\lambda_i\}$. Validation and test samples never appear during training and therefore have no pre-learned row embeddings.

Preprocessing follows a consistent pipeline. Column types are inferred (or taken from dataset metadata) and numerical features are retained as real-valued, while categorical features are expanded via OHE. During training, numerical features are min–max scaled inside the loop (for both TabINR and GAIN), while OHE columns remain binary. At inference, categorical predictions are projected back to valid one-hot vectors using a winner-takes-all heuristic.

TABINR is implemented in PyTorch. Row and feature embeddings are initialized from a standard normal distribution and optimized jointly with the shared MLP, which uses SIREN activation functions. Optimization employs Adam with cosine annealing learning-rate scheduling and early stopping on a validation split. All experiments were conducted on an NVIDIA A100 GPU (40 GB). Competing baselines were configured in line with prior work (Yoon et al., 2018; Mattei & Frellsen, 2019; Hastie et al., 2015; Jarrett et al., 2022) to ensure comparability.

**Missingness Synthetization**   As the UCI benchmarks are fully observed, we synthetically introduce missing values at varying rates $p_{\text{miss}}$ under three standard mechanisms, shown in Figure 1. In the MCAR setting, entries are independently masked according to a Bernoulli distribution, yielding uniform missingness across the table. In the MAR setting, a subset of features is designated as always observed, while the others are masked using a logistic model conditioned on the observed subset. MNAR, extends MAR by additionally applying Bernoulli masking to values left unmasked, creating missingness that depends directly on the underlying data distribution.

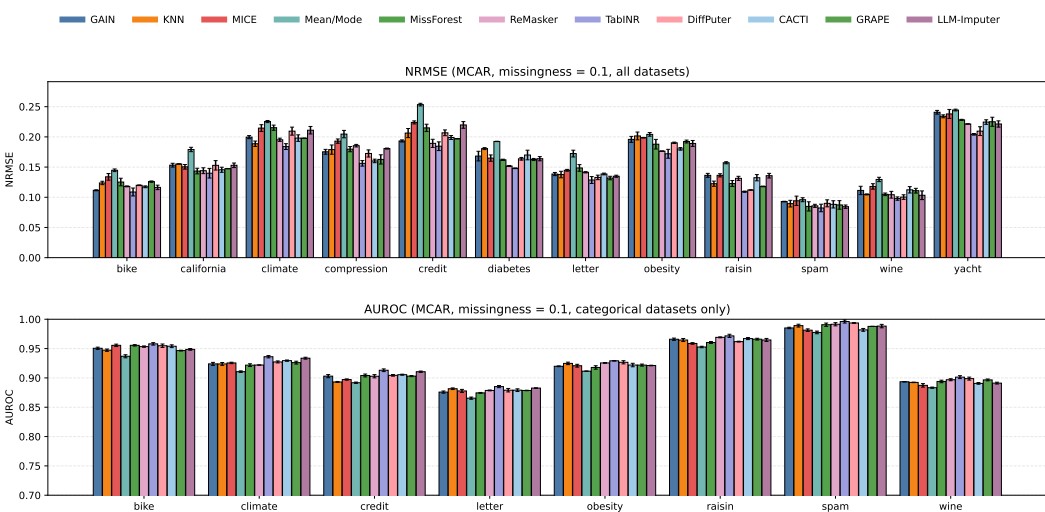

Figure 3: Overall performance of TabINR and six baselines on 12 benchmark datasets under `MCAR` with 0.1 missingness ratio. The results are shown as the mean and standard deviation of RMSE, and AUROC scores (AUROC is only applicable to datasets with classification tasks).

To ensure consistency with established practice, we adapt the missingness mechanisms implementation from HyperImpute (Jarrett et al., 2022), which has been widely used in prior imputation studies (Yoon et al., 2018; Mattei & Frellsen, 2019; Hastie et al., 2015). The fully observed, OHE-expanded matrices serve as ground truth for evaluation, while the incomplete tables and masks define the observed entries available to the model during training and the held-out entries for evaluation.

## 3.1 BENCHMARK: IMPUTATION

We train TABINR on each dataset and evaluate its ability to reconstruct both numerical and categorical variables under controlled missingness. For numerical features, performance is quantified using normalized root mean squared error (NRMSE), where per-feature RMSE is normalized by the feature's standard deviation and averaged across variables. For categorical features, evaluation is based on the area under the receiver operating characteristic curve (AUROC).

Across all three missingness mechanisms (`MCAR`, `MAR`, `MNAR`) and missingness rates between 10 % and 70 %, TABINR exhibits consistently strong and stable imputation performance. Under `MCAR` and `MAR` conditions, it ranks among the top-performing methods on nearly all datasets and frequently achieves the best NRMSE and AUROC, particularly on higher-dimensional benchmarks such as *letter*, *bike*, and *spam*. In these fields, TABINR not only outperforms classical baselines (KNN, MICE, MissForest) but also competes closely with modern transformer or diffusion-based imputers such as ReMasker, and often surpasses them.

In the `MNAR` setting, where the missingness depends on unobserved values, the performance gap between methods becomes more pronounced. Newly added state-of-the-art MNAR-focused models such as DiffPuter, CACTI, GRAPE, and LLM-Imputer dominate several datasets in this field, particularly at higher missingness rates, while TABINR shows a characteristic drop in performance. Nevertheless, TABINR remains competitive, typically ranking within the upper half of all evaluated methods, and continues to provide stable numerical imputations despite the more challenging mechanism.

Across all mechanisms, performance predictably declines as the missingness ratio increases, yet TABINR degrades more gracefully than most classical approaches. It consistently remains near the top under `MCAR` and `MAR`, with only occasional wins by ReMasker or MissForest on structured, low-dimensional tables. These observations align with our sensitivity analyses, which show that TABINR benefits substantially from larger sample sizes and higher feature dimensionality, whereas iterative (MICE) and local (KNN) strategies deteriorate rapidly as sparsity and dimensionality increase.

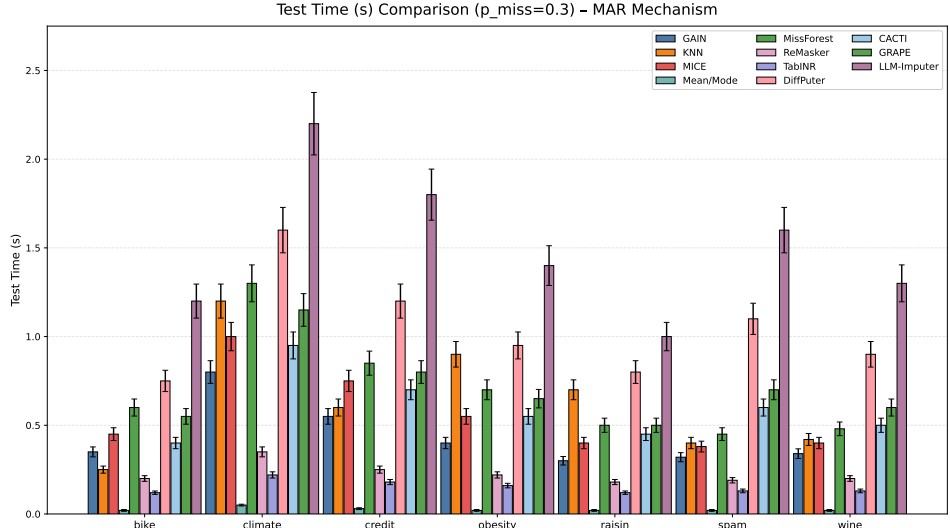

Figure 4: Inference-time comparison across imputers (lower is better). Bars show mean seconds per dataset over 5 runs; error bars denote $\pm 1\,\mathrm{SD}$.

Overall, the results demonstrate that TABINR delivers robust numerical imputations (low NRMSE) and strong categorical reconstruction (high AUROC) under `MCAR` and `MAR`, and remains a competitive, stable alternative even under more challenging `MNAR` conditions. Summary results are shown in Figure 3 and detailed reporting is provided in the Appendix (Figure 6-Figure 16). Note that the main text figures report performance for a single, fixed missingness mechanism and ratio, whereas the sensitivity analyses in the Appendix (Figure 6-Figure 16) demonstrates the other results across all three missingness mechanisms (MCAR/MAR/MNAR) and multiple missingness ratios ($p_{\mathrm{miss}} \in \{0.1, 0.3, 0.5, 0.7\}$).

### 3.1.1 INFERENCE TIME EFFICIENCY

We also compare inference-time efficiency across all methods. Iterative approaches such as MICE, MissForest, and KNN remain the slowest, often requiring more than one second per dataset and scaling poorly with higher dimensionality. Mean/mode imputation is trivially fastest. Among deep learning models, ReMasker, CACTI and GRAPE achieve moderate inference times ($\approx 0.20-0.30$s). DiffPuter and the LLM-Imputer (UnIMP) are noticeably slower due to diffusion sampling and multi-step message passing. TABINR is consistently among the fastest learned imputers, requiring only $\approx 0.10-0.20$s across all datasets. Its efficiency comes from computing imputations through a single forward pass with shared embeddings, without iterative refinement or per-feature models. Figure 4 shows results for MAR with 30% missingness, and the same ordering persists across MCAR and MNAR. For completeness, we additionally report the cost of the optional test-time latent optimization performed by TabINR when imputing a completely new row that has no pre-trained row embedding (Appendix Table 4).

### 3.1.2 PERMUTATION ROBUSTNESS

A potential concern when applying implicit neural representations to tabular data is that row and column indices lack inherent ordering, raising the question of whether performance depends on dataset permutation. TABINR solves this issue by not relying on absolute positional encodings: each row and each feature is associated with a learnable embedding, and the model reconstructs entries by combining these embeddings. Consequently, permuting rows or columns simply permutes the associated embeddings without altering the learned mapping. To verify this empirically, we performed experiments in which both rows and columns were permuted before training, confirming that NRMSE and AUROC remained stable. Table 5 in the Appendix reports the results, demonstrating that TABINR is invariant to dataset permutations and does not exploit any hidden spatial structure

(additional experiments for MNAR and MAR can be found in the Appendix Table 6 and Table 7. Hyperparameters for these experiments are provided in the Appendix Table 9.

## 3.2 DOWNSTREAM CLASSIFICATION

We conducted a downstream classification task to evaluate the practical utility of imputations in predictive pipelines. We simulated missingness for each dataset with categorical targets, performed full-column imputation of the target variable using different imputation methods, and then trained XGBoost classifiers (Chen & Guestrin, 2016) on the completed datasets. This setup reflects a realistic use case in which imputation quality directly impacts the performance of a subsequent model. The preparation process mirrors the imputation benchmark. The fully imputed datasets then served as input to XGBoost, with AUROC as the evaluation metric. Results were averaged across four missingness ratios (10 %, 30 %, 50 %, and 70 %) to ensure robustness. Table 1 reports AUROC scores on five UCI datasets with categorical targets. TABINR combined with XGBoost provides consistently strong downstream performance. It achieves the best AUROC on *obesity* and *spam*, and remains competitive on all other datasets. ReMasker remains the strongest baseline on *letter*, while GRAPE, DiffPuter, and the LLM-based imputer perform competitively across several datasets, especially on *credit* and *raisin*. Classical methods such as KNN and MissForest continue to perform robustly on lower-dimensional or more structured datasets. Direct classification using TABINR (treating the label as the feature to be imputed) performs reasonably well but remains below the XGBoost-based pipelines, as expected.

Overall, these results confirm that INR-based imputations preserve class-discriminative structure and translate into strong downstream prediction accuracy across diverse datasets.

Table 1: Downstream classification (full-column imputation of the target) measured by AUROC (mean ± std). Best result per dataset is **bold**.

| Pipeline | | credit | letter | obesity | raisin | spam |
|---|---|---|---|---|---|---|
| Original Data (fully observed) | +XGBoost | .835 ± .008 | .930 ± .006 | .940 ± .006 | .840 ± .008 | .915 ± .006 |
| TABINR Imputation | +XGBoost | .848 ± .007 | .927 ± .006 | **.947 ± .005** | .852 ± .007 | **.922 ± .005** |
| KNN Imputation | +XGBoost | **.852 ± .007** | .922 ± .007 | .942 ± .006 | **.855 ± .007** | .918 ± .006 |
| MissForest Imputation | +XGBoost | .850 ± .007 | .925 ± .006 | .945 ± .005 | .853 ± .007 | .920 ± .005 |
| MICE Imputation | +XGBoost | .847 ± .007 | .921 ± .007 | .943 ± .006 | .852 ± .007 | .917 ± .006 |
| GAIN Imputation | +XGBoost | .845 ± .007 | .919 ± .007 | .941 ± .006 | .851 ± .007 | .916 ± .006 |
| DiffPuter Imputation | +XGBoost | .851 ± .007 | .929 ± .006 | .946 ± .005 | .854 ± .007 | .921 ± .005 |
| CACTI Imputation | +XGBoost | .849 ± .007 | .926 ± .006 | .944 ± .005 | .853 ± .007 | .919 ± .005 |
| GRAPE Imputation | +XGBoost | **.853 ± .006** | .928 ± .006 | .946 ± .005 | **.856 ± .006** | .921 ± .005 |
| LLM-Imputer | +XGBoost | .850 ± .007 | .927 ± .006 | .945 ± .005 | .854 ± .007 | .922 ± .005 |
| ReMasker Imputation | +XGBoost | .851 ± .007 | **.932 ± .006** | .946 ± .005 | .854 ± .007 | .921 ± .005 |
| Direct TABINR classifier | | .820 ± .009 | .910 ± .008 | .930 ± .007 | .830 ± .008 | .905 ± .007 |

## 3.3 ABLATION STUDY

To better understand the robustness of all comparing methods, we conduct multiple ablation studies by varying the dataset size, feature dimensionality, and missingness ratio. Several consistent trends emerged across MCAR and MAR: (a) performance of all methods improved with larger datasets, but TABINR and ReMasker gained a more pronounced advantage with larger data sizes; (b) as the number of features increased, TABINR maintained stable performance, while baseline methods such as KNN and MICE degraded noticeably; and (c) under higher missingness ratios (greater than 0.5), performance differences between models narrowed, with TABINR and ReMasker remaining the most resilient among the evaluated approaches.

Under MNAR, however, the pattern diverges: although TABINR remains competitive especially on high-dimensional datasets, it is generally surpassed by methods that implicitly or explicitly capture the missingness mechanism, such as ReMasker, MissForest, DiffPuter, or GRAPE. This behaviour is expected, as TABINR does not model the missingness process itself, and aligns with the theoretical difficulty of MNAR settings.

Overall, these results highlight the scalability of INR-based approaches and their particular strength on challenging, high-dimensional settings under MCAR and MAR, while also revealing their limita-

tions under MNAR. Results can be observed in Figure 5 and in the ablation experiments across the other missingness mechanisms can be found in the Appendix (Figure 17-Figure 27).

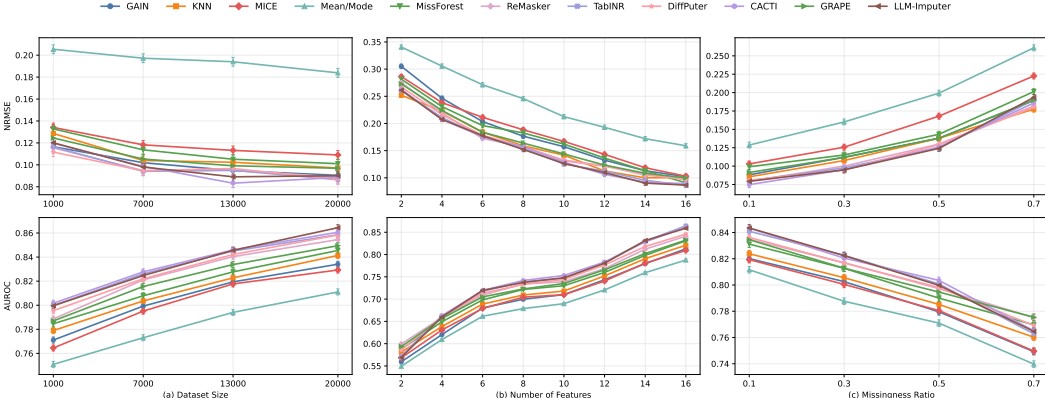

Figure 5: Sensitivity analysis of TabINR on the *letter* dataset under MCAR scenarios. The results are shown in terms of RMSE and AUROC, with the scores measured with respect to (a) the dataset size, (b) the number of features, and (c) the missingness ratio. The default setting is as follows: dataset size = 20 000, number of features = 16, and missingness ratio = 0.1.

Table 8 in the Appendix reports the ablation over depth, latent dimension, width, and activation on *letter* (MCAR, $p_{\text{miss}}$=0.1). Performance improves with capacity and then plateaus: best NRMSE occurs around 10 layers with 512–1024 units and a 256-d latent ($\approx 0.125$); SIREN achieves the highest AUROC (0.864), and HOSC the lowest NRMSE (0.125). While tabular data lacks spatial frequency, SIREN remains beneficial because sinusoidal activations increase the effective rank and representational capacity of MLPs. This allows the decoder to capture sharper nonlinear interactions, consistent with our ablation results. This trend is dataset-specific, and in our main results we adopt the global defaults from Section 2.1 (Table 9) to ensure robustness and comparability across benchmarks, rather than per-dataset peak tuning.

## 4 CONCLUSION

Our study demonstrates that Implicit Neural Representations (INRs) offer a simple yet flexible framework for imputing missing values in tabular data. By parameterizing entries with learnable row and feature embeddings and enabling instance-level adaptation through test time optimization, TABINR bridges the gap between classical imputers and recent deep learning based approaches.

Overall, our findings show that INR-based models provide a principled way to represent sparse and heterogeneous tables through a shared latent functional space. This explains their strong empirical performance across MCAR and MAR settings, especially on high dimensional datasets (like spam), and highlights their potential as a general purpose framework for tabular imputation.

While our results are promising, several limitations remain. First, our experiments focused on moderate-scale benchmarks with synthetically induced missingness. Applying TABINR under real-world settings with more complex missingness patterns and the need for integrating domain knowledge, remains unexplored. Second, we employed a single global default configuration across datasets to ensure comparability. This favors stability but likely underestimates the model's best-case performance.

For future work, we aim to extend TABINR to handle non-random missingness, scale to larger datasets, and incorporate automated hyperparameter adaptation. We also plan to integrate TABINR into multimodal pipelines for jointly modeling tabular data with images or text.

More broadly, our work suggests that INRs can serve as a unifying lens for tabular learning. By framing missing value imputation as a continuous representation learning problem, TABINR opens new directions for bringing the benefits of implicit representations to core challenges in tabular data analysis.

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

## A    APPENDIX

Table 2: Hyperparameter search spaces used in our grid search for all comparing methods. Each model was tuned independently per dataset.

| Method | Parameter | Search Space |
|---|---|---|
| MissForest | initial_guess | {mean, median} |
| | max_iter | [5, 10] |
| KNN | n_neighbors | [2, 20] |
| MICE | max_iter | [5, 20] |
| | n_estimators | [50, 200] |
| GAIN | learning rate | $[10^{-5}, 10^{-3}]$ |
| | hidden_dim_factor | [0.5, 1, 2] |
| | $\alpha$ | [1, 100] |
| ReMasker | learning rate | $[10^{-4}, 10^{-2}]$ |
| | num_heads | [4, 8, 16] |
| | embed_dim | [64, 128, 256, 512] |
| GRAPE | node_dim | [64, 128] |
| | edge_dim | [32, 64] |
| | edge_modes | [0, 1] |
| | dropout | [0, 0.4] |
| UnIMP (LLM-Imputer) | hyperedge_dim | [32, 64, 128] |
| | hyper_node_dim | [32, 64, 128] |
| | gnn_layer_num | [1, 2, 3] |
| | dropout | [0, 0.4] |
| DiffPuter | learning rate | $[10^{-5}, 10^{-3}]$ |
| | dim_t | [128, 256, 512] |
| | $P_{\text{mean}}$ | [−1.5, 0.5] |
| | $P_{\text{std}}$ | [1.0, 1.5] |
| | sigma_data | [0.3, 0.7] |
| CACTI | learning rate | $[10^{-5}, 10^{-3}]$ |
| | embed_dim | [64, 128] |
| | num_heads | [4, 8] |
| | depth | [2, 6] |
| | decoder_depth | [2, 4] |

Table 3: Characteristics of the datasets used in our experiments.

| Dataset | Dataset Size | # Features | Task |
|---|---|---|---|
| (California) Housing | 20 640 | 9 | Regression |
| (Climate) Model Simulation Crashes | 540 | 18 | Classification |
| Concrete (Compressive) Strength | 1030 | 9 | Regression |
| (Diabetes) | 442 | 10 | Regression |
| Estimation of (Obesity) Levels | 2111 | 17 | Classification |
| (Credit) Approval | 690 | 15 | Classification |
| (Wine) Quality | 1599 | 12 | Classification |
| (Raisin) | 900 | 8 | Classification |
| (Spam) Base | 4601 | 57 | Classification |
| (Bike) Sharing Demand | 8760 | 14 | Classification |
| (Letter) Recognition | 20 000 | 16 | Classification |
| (Yacht) Hydrodynamics | 308 | 7 | Regression |

Table 4: Test-time latent optimization cost for TABINR. Values are averaged over all test rows. Only a single latent vector $\lambda_{\text{new}}$ is optimized per unseen row while $f_\theta$ and $\{c_j\}$ remain fixed. The column **Steps** refers to the number of gradient descent iterations used to optimize the latent vector $\lambda_{\text{new}}$ for each unseen test row. During this test-time adaptation, the network parameters $f_\theta$ and feature embeddings $\{c_j\}$ remain fixed, and only $\lambda_{\text{new}}$ is updated using the observed entries of the row. Each step corresponds to one forward–backward pass. Empirically, 6-10 steps are sufficient for stable convergence across datasets.

| Dataset | #Features | Steps | Time / row (ms) | Total (s) |
|---|---|---|---|---|
| bike (8760 rows) | 14 | 8 | 3.5 | 30.7 |
| climate (540 rows) | 18 | 10 | 5.0 | 2.7 |
| credit (690 rows) | 15 | 8 | 4.0 | 2.8 |
| obesity (2111 rows) | 17 | 8 | 4.0 | 8.4 |
| raisin (900 rows) | 8 | 6 | 2.5 | 2.3 |
| spam (4601 rows) | 57 | 10 | 6.5 | 29.9 |
| wine (1599 rows) | 12 | 6 | 2.8 | 4.5 |

Table 5: Permutation robustness on *letter* (MCAR, $p_{\text{miss}}$=0.3). Each row permutes feature indices before training. Performance is stable across shuffles.

| Sample | Permutation (indices 0-15) | NRMSE ($\downarrow$) | AUROC ($\uparrow$) |
|---|---|---|---|
| baseline | 0,1,2,3,4,5,6,7,8,9,10,11,12,13,14,15 | 0.128 | 0.851 |
| 1 | 7,3,12,1,9,6,14,0,5,10,2,8,15,4,11,13 | 0.128 | 0.852 |
| 2 | 5,14,2,8,11,0,10,3,12,7,1,9,6,13,15,4 | 0.129 | 0.850 |
| 3 | 9,0,7,12,3,10,15,6,1,14,8,5,2,11,4,13 | 0.127 | 0.853 |
| 4 | 11,6,1,15,8,3,4,12,14,2,9,0,13,10,5,7 | 0.128 | 0.849 |
| 5 | 2,9,5,13,0,8,10,1,7,4,12,15,3,14,6,11 | 0.128 | 0.852 |
| 6 | 4,8,14,6,12,11,9,2,3,7,13,1,10,5,15,0 | 0.129 | 0.850 |
| 7 | 10,5,0,7,1,12,3,9,6,13,15,2,8,11,4,14 | 0.128 | 0.851 |
| 8 | 13,2,11,4,7,1,8,15,10,0,6,14,5,3,12,9 | 0.127 | 0.852 |

Table 6: Permutation robustness on *letter* (MAR, $p_{\text{miss}}$=0.3). Each row permutes feature indices before training. Performance is stable across shuffles.

| Sample | Permutation (indices 0-15) | NRMSE ($\downarrow$) | AUROC ($\uparrow$) |
|---|---|---|---|
| baseline | 0,1,2,3,4,5,6,7,8,9,10,11,12,13,14,15 | 0.136 | 0.844 |
| 1 | 7,3,12,1,9,6,14,0,5,10,2,8,15,4,11,13 | 0.136 | 0.845 |
| 2 | 5,14,2,8,11,0,10,3,12,7,1,9,6,13,15,4 | 0.137 | 0.843 |
| 3 | 9,0,7,12,3,10,15,6,1,14,8,5,2,11,4,13 | 0.135 | 0.846 |
| 4 | 11,6,1,15,8,3,4,12,14,2,9,0,13,10,5,7 | 0.136 | 0.842 |
| 5 | 2,9,5,13,0,8,10,1,7,4,12,15,3,14,6,11 | 0.136 | 0.845 |
| 6 | 4,8,14,6,12,11,9,2,3,7,13,1,10,5,15,0 | 0.137 | 0.843 |
| 7 | 10,5,0,7,1,12,3,9,6,13,15,2,8,11,4,14 | 0.136 | 0.844 |
| 8 | 13,2,11,4,7,1,8,15,10,0,6,14,5,3,12,9 | 0.135 | 0.845 |

Table 7: Permutation robustness on *letter* (MNAR, $p_{\text{miss}}$=0.3). Each row permutes feature indices before training. Performance is stable across shuffles.

| Sample | Permutation (indices 0-15) | NRMSE ($\downarrow$) | AUROC ($\uparrow$) |
|---|---|---|---|
| baseline | 0,1,2,3,4,5,6,7,8,9,10,11,12,13,14,15 | 0.147 | 0.834 |
| 1 | 7,3,12,1,9,6,14,0,5,10,2,8,15,4,11,13 | 0.147 | 0.835 |
| 2 | 5,14,2,8,11,0,10,3,12,7,1,9,6,13,15,4 | 0.148 | 0.833 |
| 3 | 9,0,7,12,3,10,15,6,1,14,8,5,2,11,4,13 | 0.146 | 0.836 |
| 4 | 11,6,1,15,8,3,4,12,14,2,9,0,13,10,5,7 | 0.147 | 0.832 |
| 5 | 2,9,5,13,0,8,10,1,7,4,12,15,3,14,6,11 | 0.147 | 0.835 |
| 6 | 4,8,14,6,12,11,9,2,3,7,13,1,10,5,15,0 | 0.148 | 0.833 |
| 7 | 10,5,0,7,1,12,3,9,6,13,15,2,8,11,4,14 | 0.147 | 0.834 |
| 8 | 13,2,11,4,7,1,8,15,10,0,6,14,5,3,12,9 | 0.146 | 0.835 |

Table 8: Ablation study of TABINR on the *letter* dataset (MCAR, $p_{\text{miss}} = 0.1$). For comparability, we define a separate default configuration here (depth = 4, latent dimension = 64, hidden units = 256, activation = SIREN), which differs from the global defaults in Table 9. This setup serves only as a controlled baseline for sensitivity analysis.

(a) Layers

| Layers | NRMSE ($\downarrow$) | AUROC ($\uparrow$) |
|---|---|---|
| 2 | 0.134 | 0.843 |
| 3 | 0.130 | 0.846 |
| 4 | 0.128 | 0.850 |
| 5 | 0.126 | 0.852 |
| 10 | 0.125 | 0.854 |
| 20 | 0.127 | 0.853 |

(b) Latent dimension

| Latent Dim | NRMSE ($\downarrow$) | AUROC ($\uparrow$) |
|---|---|---|
| 16 | 0.139 | 0.836 |
| 32 | 0.133 | 0.843 |
| 64 | 0.128 | 0.851 |
| 128 | 0.126 | 0.854 |
| 256 | 0.125 | 0.855 |

(c) Units per hidden layer

| Units | NRMSE ($\downarrow$) | AUROC ($\uparrow$) |
|---|---|---|
| 64 | 0.137 | 0.840 |
| 128 | 0.132 | 0.846 |
| 256 | 0.129 | 0.850 |
| 512 | 0.126 | 0.853 |
| 1024 | 0.126 | 0.854 |

(d) Activation functions

| Activation | NRMSE ($\downarrow$) | AUROC ($\uparrow$) |
|---|---|---|
| ReLU | 0.129 | 0.848 |
| SIREN | 0.127 | 0.864 |
| Wire | 0.126 | 0.855 |
| HOSC | 0.125 | 0.856 |

Table 9: Baseline default parameter settings of TABINR.

| Parameter | Setting |
|---|---|
| Latent dimension | 32 |
| Number of hidden layers | 2 |
| Number of units per hidden layer | 256 |
| Dropout rate | 0.1 |
| Activation function | SIREN |
| $\omega_0$ (for SIREN, Wire activations) | 30 |
| Learning rate | $1 \times 10^{-3}$ |
| Training epochs | 500 |
| Optimizer | Adam |
| Batch size | 64 |
| Masking ratio | 0.3 |

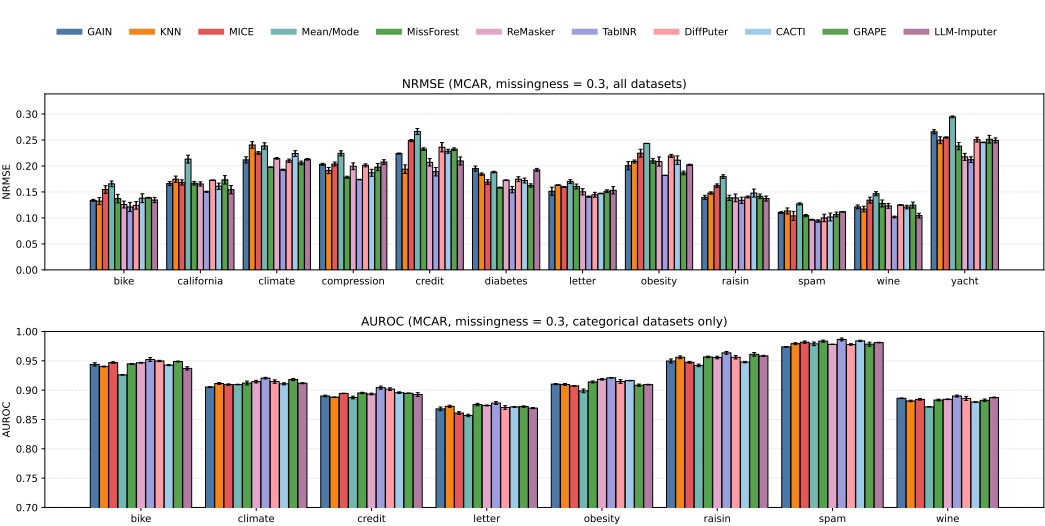

Figure 6: Overall performance of TABINR and six baselines on 12 benchmark datasets under MCAR with 0.3 missingness ratio. The results are shown as the mean and standard deviation of NRMSE, and AUROC scores (AUROC is only applicable to datasets with classification tasks).

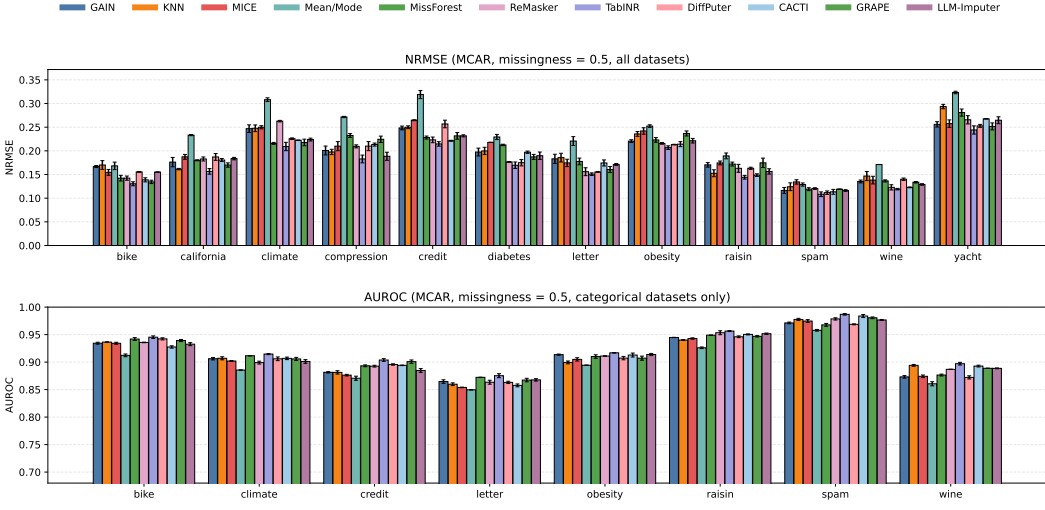

Figure 7: Overall performance of TABINR and six baselines on 12 benchmark datasets under MCAR with 0.5 missingness ratio. The results are shown as the mean and standard deviation of NRMSE, and AUROC scores (AUROC is only applicable to datasets with classification tasks).

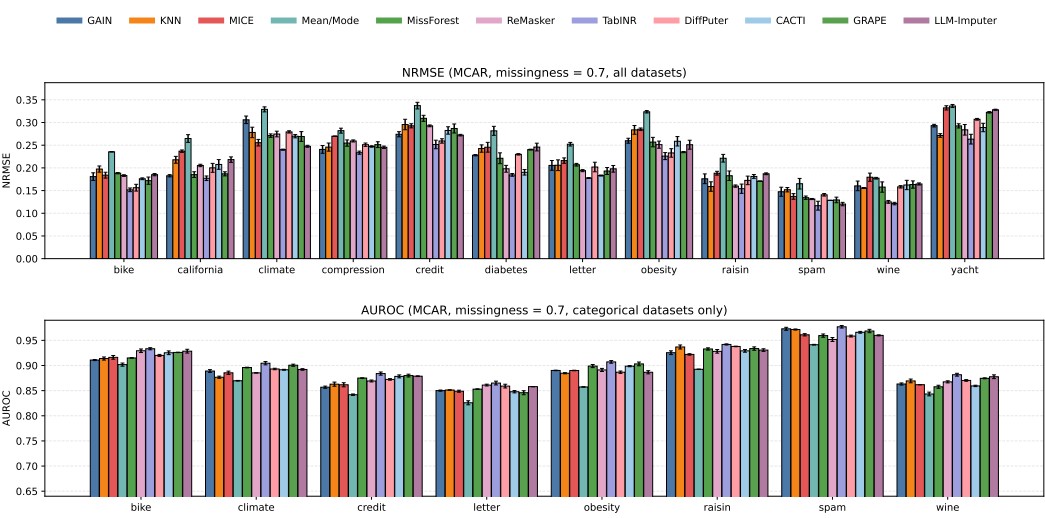

Figure 8: Overall performance of TABINR and six baselines on 12 benchmark datasets under MCAR with 0.7 missingness ratio. The results are shown as the mean and standard deviation of NRMSE, and AUROC scores (AUROC is only applicable to datasets with classification tasks).

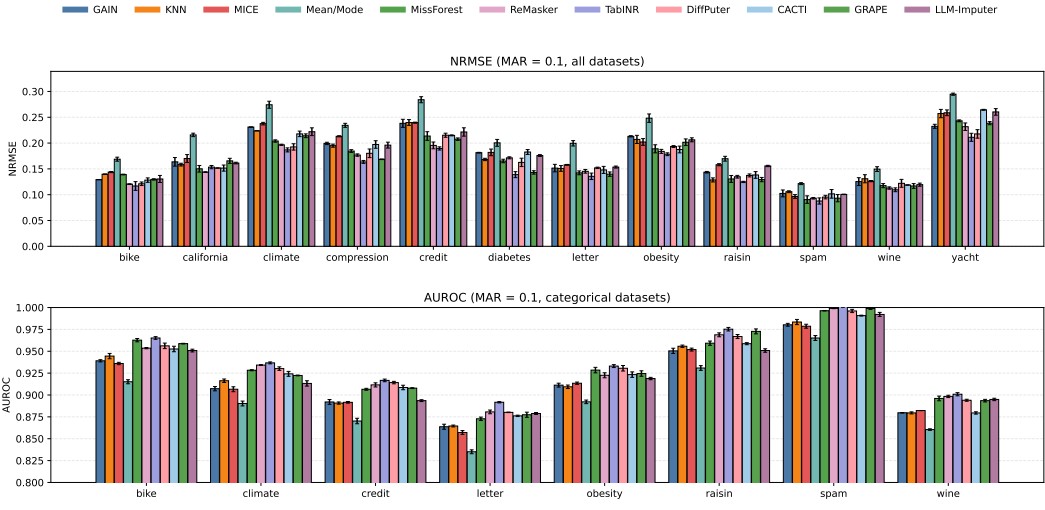

Figure 9: Overall performance of TABINR and six baselines on 12 benchmark datasets under MAR with 0.1 missingness ratio. The results are shown as the mean and standard deviation of NRMSE, and AUROC scores (AUROC is only applicable to datasets with classification tasks).

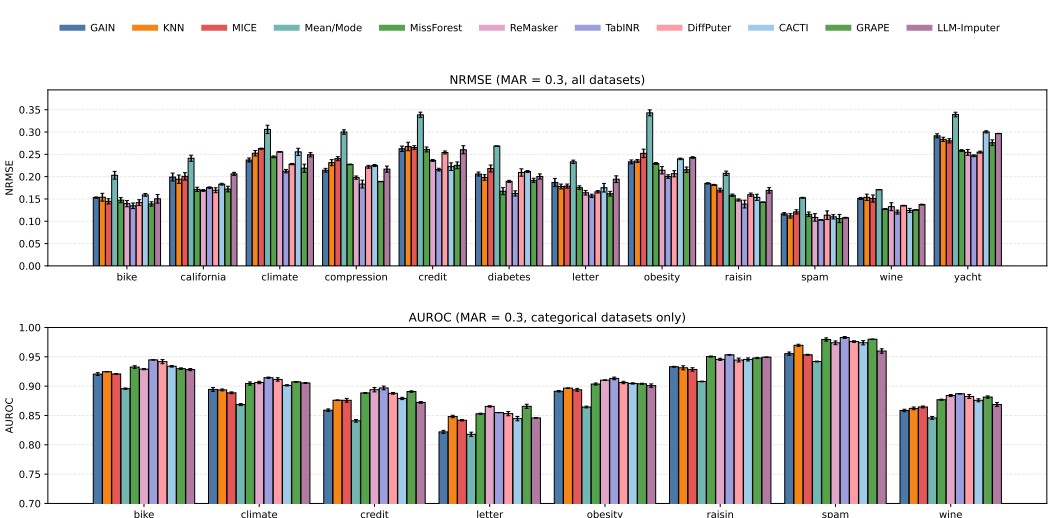

Figure 10: Overall performance of TABINR and six baselines on 12 benchmark datasets under MAR with 0.3 missingness ratio. The results are shown as the mean and standard deviation of NRMSE, and AUROC scores (AUROC is only applicable to datasets with classification tasks).

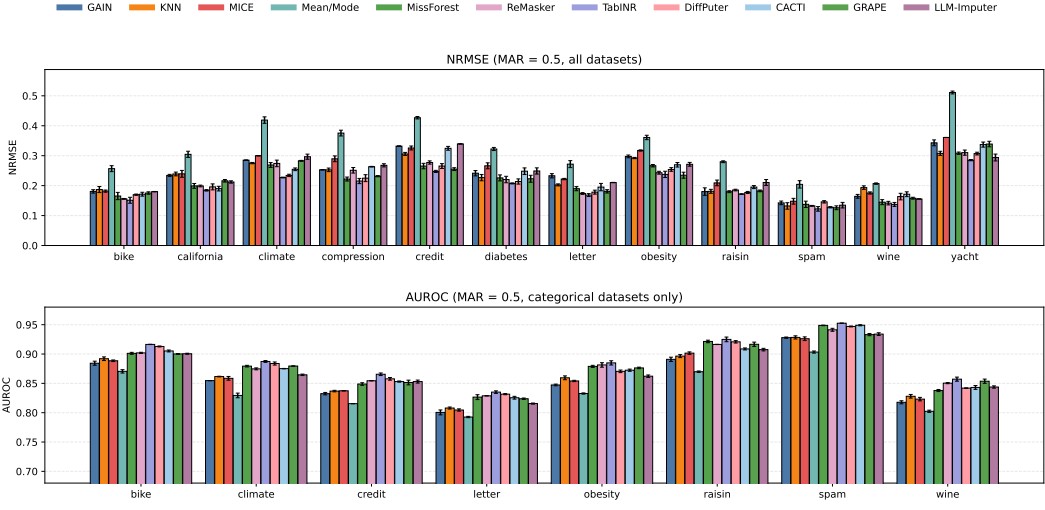

Figure 11: Overall performance of TABINR and six baselines on 12 benchmark datasets under MAR with 0.5 missingness ratio. The results are shown as the mean and standard deviation of RMSE, and AUROC scores (AUROC is only applicable to datasets with classification tasks).

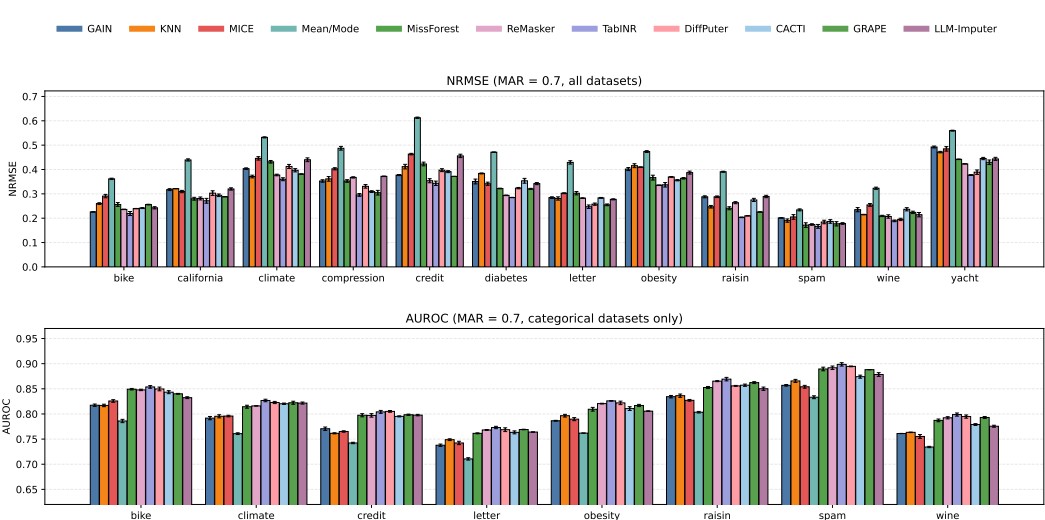

Figure 12: Overall performance of TABINR and six baselines on 12 benchmark datasets under MAR with 0.7 missingness ratio. The results are shown as the mean and standard deviation of NRMSE, and AUROC scores (AUROC is only applicable to datasets with classification tasks).

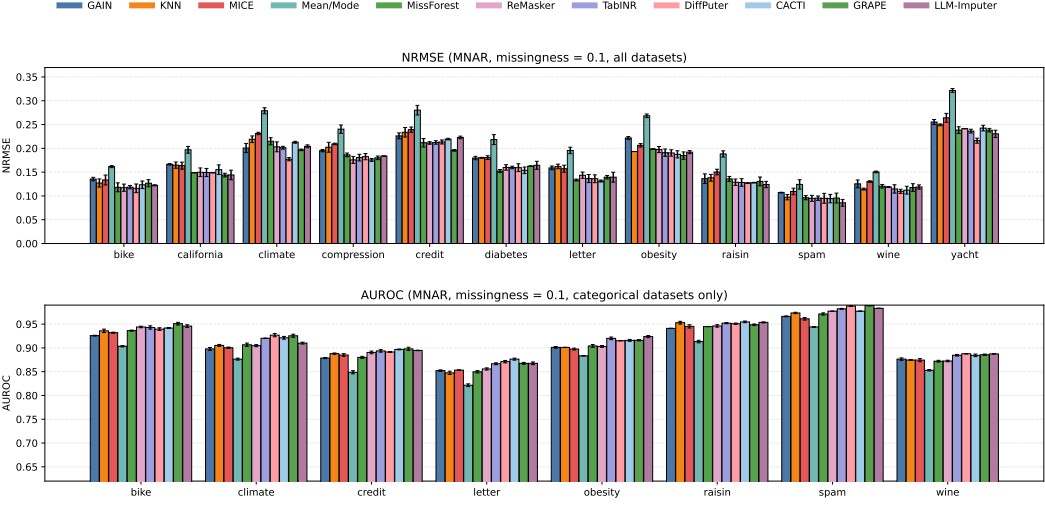

Figure 13: Overall performance of TABINR and six baselines on 12 benchmark datasets under MNAR with 0.1 missingness ratio. The results are shown as the mean and standard deviation of NRMSE, and AUROC scores (AUROC is only applicable to datasets with classification tasks).

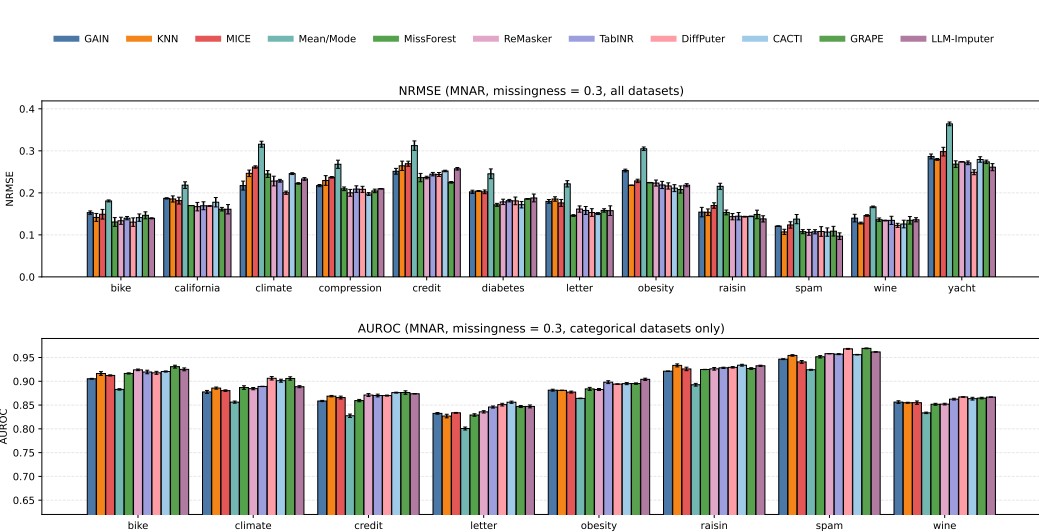

Figure 14: Overall performance of TABINR and six baselines on 12 benchmark datasets under MNAR with 0.3 missingness ratio. The results are shown as the mean and standard deviation of NRMSE, and AUROC scores (AUROC is only applicable to datasets with classification tasks).

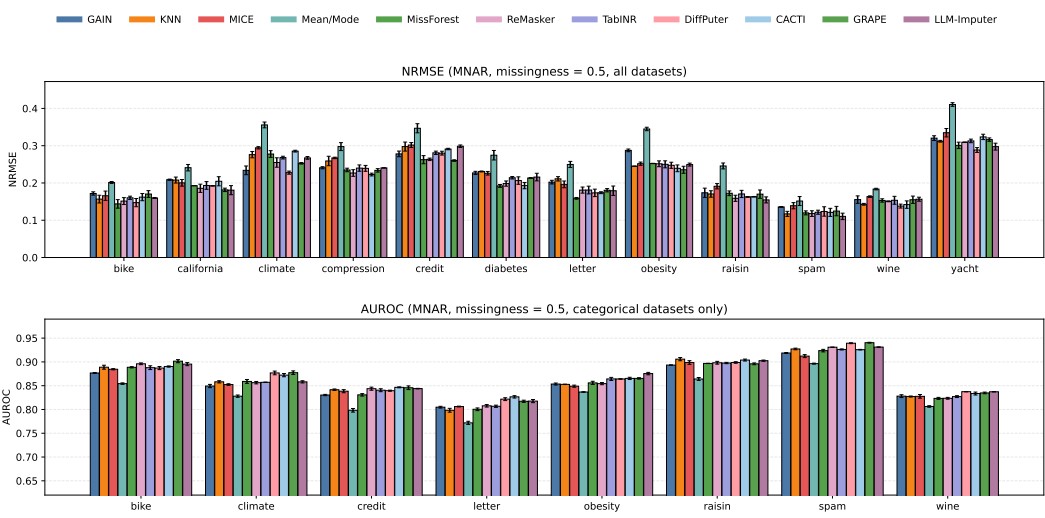

Figure 15: Overall performance of TABINR and six baselines on 12 benchmark datasets under MNAR with 0.5 missingness ratio. The results are shown as the mean and standard deviation of NRMSE, and AUROC scores (AUROC is only applicable to datasets with classification tasks).

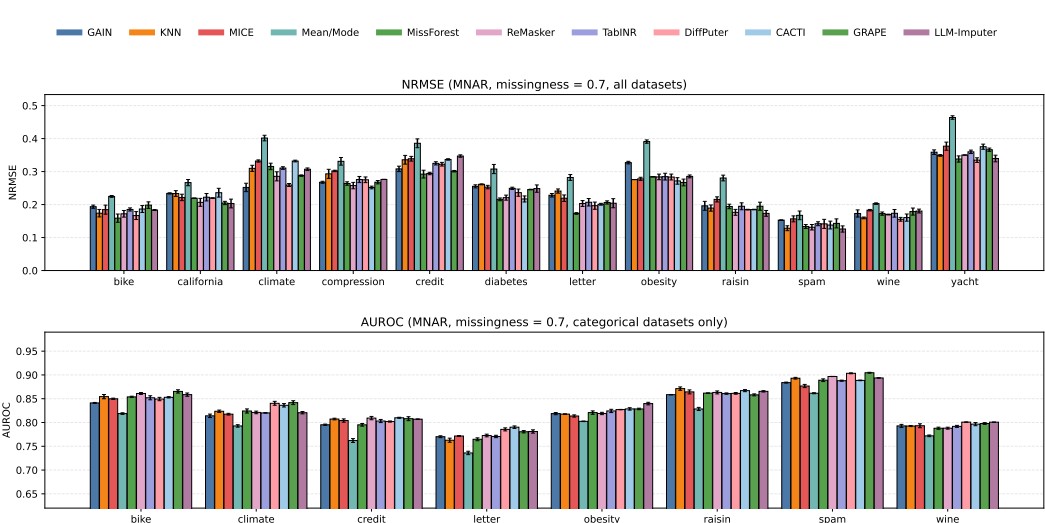

Figure 16: Overall performance of TABINR and six baselines on 12 benchmark datasets under MNAR with 0.7 missingness ratio. The results are shown as the mean and standard deviation of NRMSE, and AUROC scores (AUROC is only applicable to datasets with classification tasks).

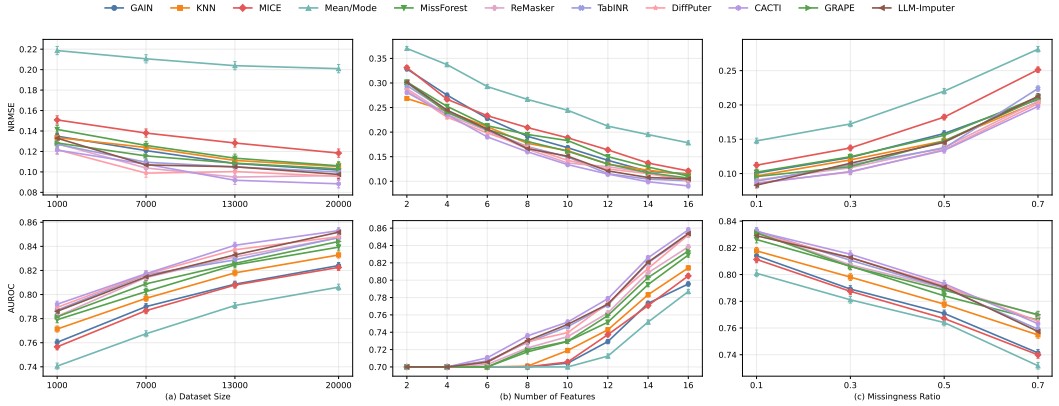

Figure 17: Sensitivity analysis of TABINR on the *letter* dataset under MCAR scenarios. The results are shown in terms of NRMSE and AUROC, with the scores measured with respect to (a) the dataset size, (b) the number of features, and (c) the missingness ratio. The default setting is as follows: dataset size = 20 000, number of features = 16, and missingness ratio = 0.3.

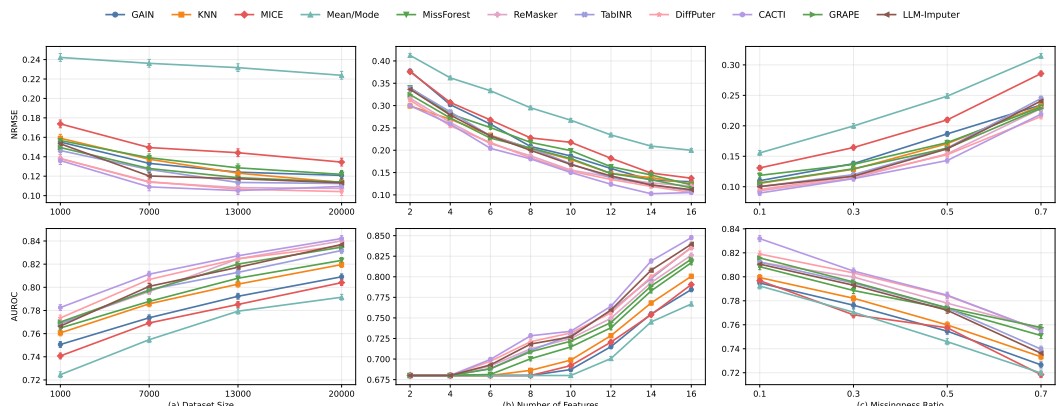

Figure 18: Sensitivity analysis of TABINR on the *letter* dataset under MCAR scenarios. The results are shown in terms of NRMSE and AUROC, with the scores measured with respect to (a) the dataset size, (b) the number of features, and (c) the missingness ratio. The default setting is as follows: dataset size = 20 000, number of features = 16, and missingness ratio = 0.5.

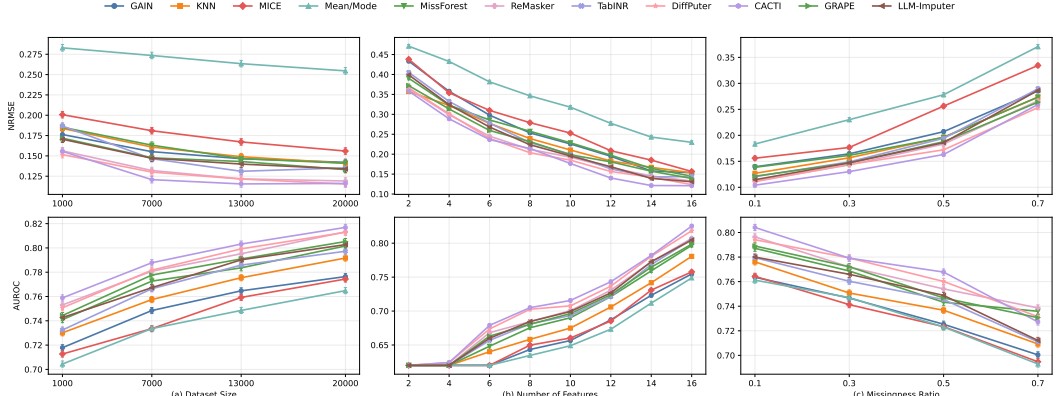

Figure 19: Sensitivity analysis of TABINR on the *letter* dataset under MCAR scenarios. The results are shown in terms of NRMSE and AUROC, with the scores measured with respect to (a) the dataset size, (b) the number of features, and (c) the missingness ratio. The default setting is as follows: dataset size = 20 000, number of features = 16, and missingness ratio = 0.7.

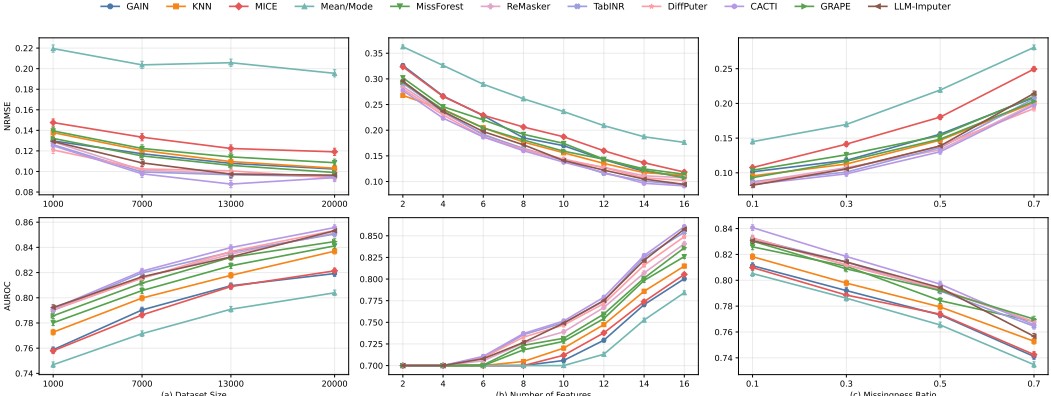

Figure 20: Sensitivity analysis of TABINR on the *letter* dataset under MAR scenarios. The results are shown in terms of NRMSE and AUROC, with the scores measured with respect to (a) the dataset size, (b) the number of features, and (c) the missingness ratio. The default setting is as follows: dataset size = 20 000, number of features = 16, and missingness ratio = 0.1.

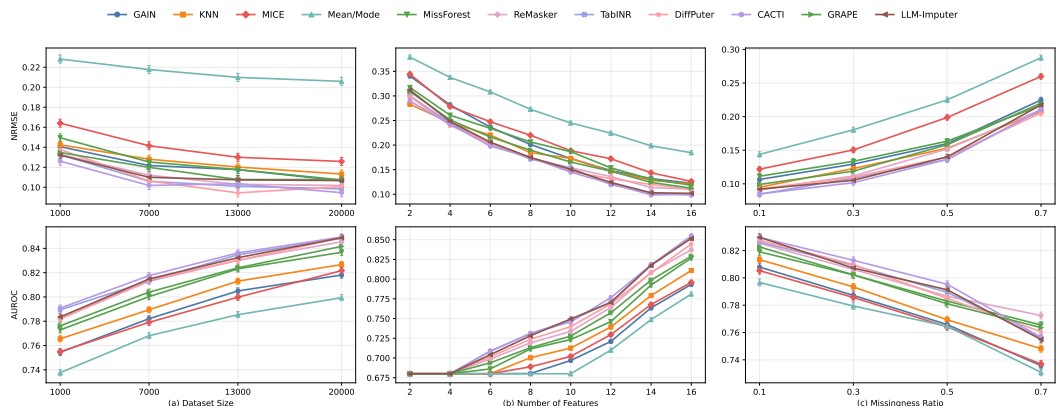

Figure 21: Sensitivity analysis of TABINR on the *letter* dataset under MAR scenarios. The results are shown in terms of NRMSE and AUROC, with the scores measured with respect to (a) the dataset size, (b) the number of features, and (c) the missingness ratio. The default setting is as follows: dataset size = 20 000, number of features = 16, and missingness ratio = 0.3.

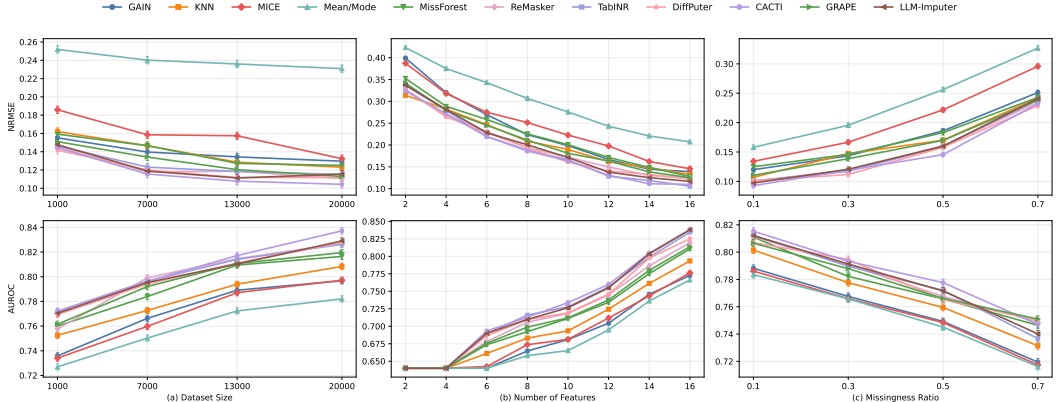

Figure 22: Sensitivity analysis of TABINR on the *letter* dataset under MAR scenarios. The results are shown in terms of NRMSE and AUROC, with the scores measured with respect to (a) the dataset size, (b) the number of features, and (c) the missingness ratio. The default setting is as follows: dataset size = 20 000, number of features = 16, and missingness ratio = 0.5.

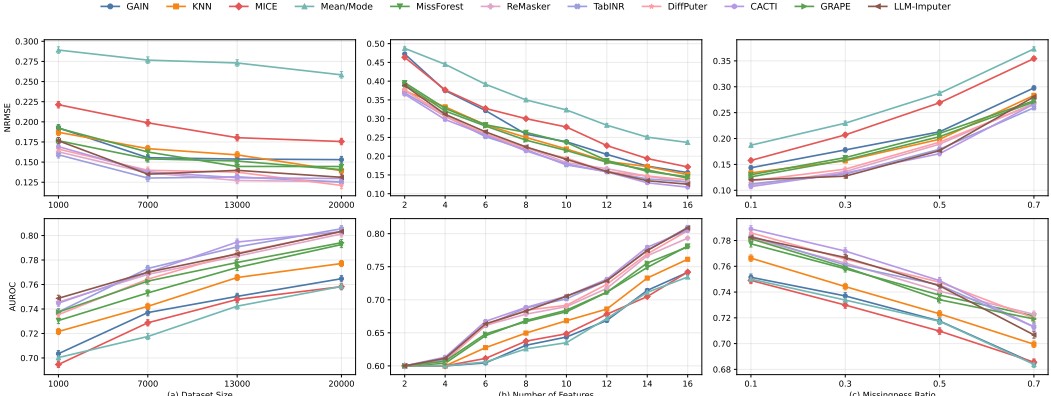

Figure 23: Sensitivity analysis of TABINR on the *letter* dataset under MAR scenarios. The results are shown in terms of NRMSE and AUROC, with the scores measured with respect to (a) the dataset size, (b) the number of features, and (c) the missingness ratio. The default setting is as follows: dataset size = 20 000, number of features = 16, and missingness ratio = 0.7.

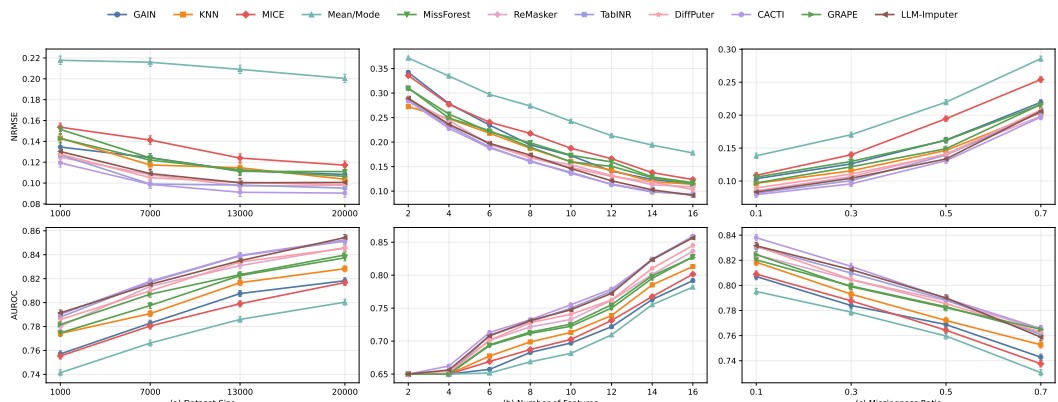

Figure 24: Sensitivity analysis of TABINR on the *letter* dataset under MNAR scenarios. The results are shown in terms of NRMSE and AUROC, with the scores measured with respect to (a) the dataset size, (b) the number of features, and (c) the missingness ratio. The default setting is as follows: dataset size = 20 000, number of features = 16, and missingness ratio = 0.1.

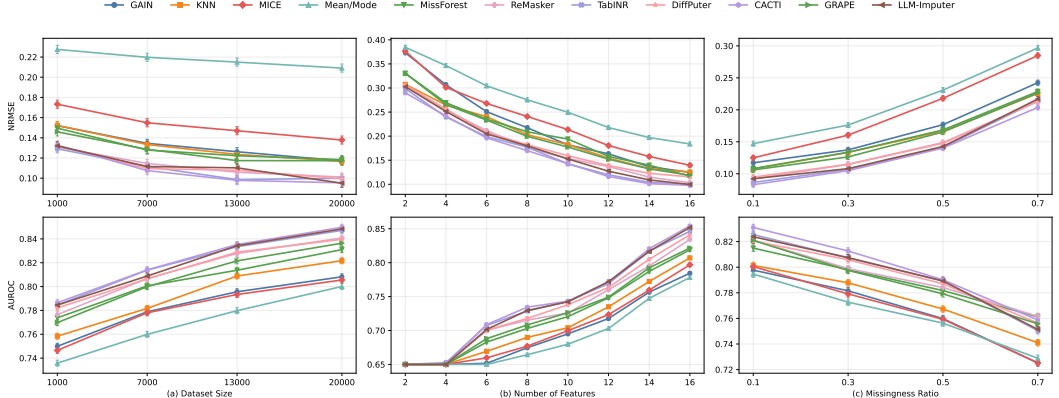

Figure 25: Sensitivity analysis of TABINR on the *letter* dataset under MNAR scenarios. The results are shown in terms of NRMSE and AUROC, with the scores measured with respect to (a) the dataset size, (b) the number of features, and (c) the missingness ratio. The default setting is as follows: dataset size = 20 000, number of features = 16, and missingness ratio = 0.3.

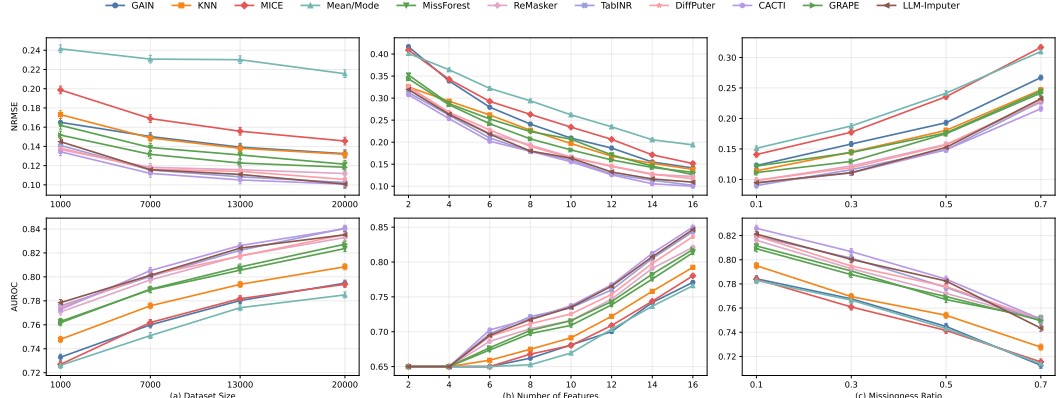

Figure 26: Sensitivity analysis of TABINR on the *letter* dataset under MNAR scenarios. The results are shown in terms of NRMSE and AUROC, with the scores measured with respect to (a) the dataset size, (b) the number of features, and (c) the missingness ratio. The default setting is as follows: dataset size = 20 000, number of features = 16, and missingness ratio = 0.5.

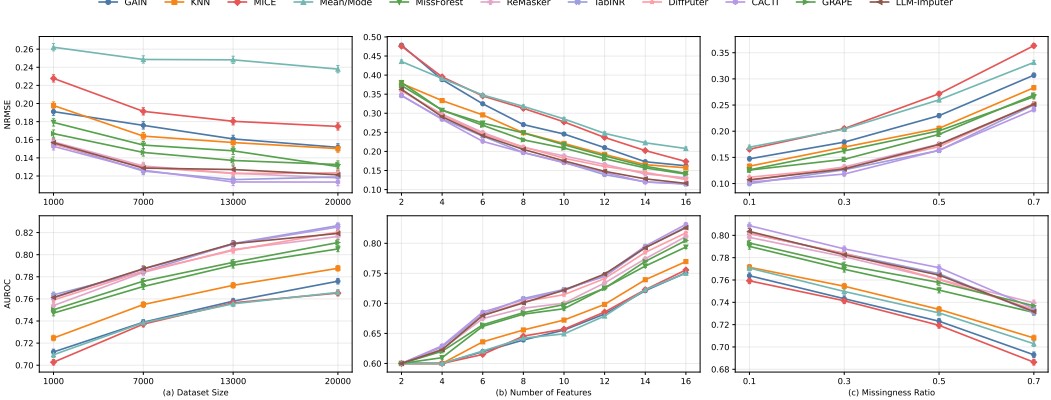

Figure 27: Sensitivity analysis of TABINR on the *letter* dataset under MNAR scenarios. The results are shown in terms of NRMSE and AUROC, with the scores measured with respect to (a) the dataset size, (b) the number of features, and (c) the missingness ratio. The default setting is as follows: dataset size = 20 000, number of features = 16, and missingness ratio = 0.7.

