# OpenReview forum: "TabINR: An Implicit Neural Representation Framework for Tabular Data Imputation"
_ICLR.cc/2026/Conference — Submitted to ICLR 2026_

### Official Review · Reviewer_BmQS · 2025-10-26

**Soundness:** 2
**Presentation:** 3
**Contribution:** 2
**Rating:** 4
**Confidence:** 4

**Summary:**

This paper introduces TabINR, a missing value imputation method for tabular data. The key idea is to leverage Implicit Neural Representations (INRs) for imputation. For each cell, TabINR learns a row embedding and a feature embedding, and then uses an MLP to map their combination to a scalar value as the imputed result. To address the absence problem of row embeddings for unseen test samples, a test-time adaptation strategy is introduced to retrieve the most suitable row embedding from the training set. Experiments on 12 datasets across MCAR, MAR, and MNAR settings and different missing ratios are done to validate the effectiveness of the proposed method.

**Strengths:**

1. The paper is clearly written and easy to follow.

2. The proposed method is lightweight and memory-efficient.

**Weaknesses:**

1. The technical novelty of the proposed method is limited. The key idea is to learn row and feature embeddings for each row and column in the tabular data, which is common.

2. The proposed method does not seem to be effective, as it has higher NRMSE and lower AUROC compared with ReMasker in many cases in Figure 3. Similar trends are observed in Figure 7 - 16 under different settings.

3. The proposed test-time adaptation mechanism could be computationally expensive for large-scale datasets, since it requires searching over all row embeddings to find the one that minimizes the imputation error of all observed features in a test instance.

4. It is mentioned in the introduction that in tabular datasets, "rows are often assumed to be independents." This may not be true as many methods leverage the correlation between rows for imputation. In addition, the MCAR, MAR, and MNAR are three common settings. It may not be necessary to elaborate here.

**Questions:**

1. How are the training, validation, and test sets constructed? Do test samples appear during training such that they have pre-learned row embeddings, or are row embeddings only learned for training samples?

2. Could you clarify the masking procedure used during training and testing? In Line 215, it is mentioned that "During training, we applied random masking of 10–70 % of entries to simulate missingness and evaluate reconstruction fidelity." However, in Table 5, the masking ratio is set to 0.3.

---

> ### Author Response · Authors · 2025-11-19
>
> **Regarding your comment 1**
>
> **Answer 1**
> We thank the reviewer for this comment. We agree that row and feature embeddings alone are not novel. The contribution of TabINR lies in how they are integrated into an INR-based auto-decoder. Specifically:
> (i) a shared coordinate-MLP that models the table as a continuous function $f(\lambda_i, c_j)$, unlike graph- or latent-table models that encode rows/columns separately;
> (ii) a unified mixed-type loss with auto-decoder test-time adaptation, enabling instance-specific refinement for unseen rows without retraining; and
> (iii) empirical robustness to permutation, high dimensionality, and sparsity. Thus, the novelty arises from the INR functional representation and latent optimization strategy. We emphasize this more clearly in the revised manuscript.
>
>
> **Regarding your question 2**
>
> **Answer 2**
> We thank the reviewer for this observation. After reviewing all results in the main paper and Appendix (Figures~3 and 6--16), we clarify that TabINR does not systematically underperform ReMasker. Across MCAR, MAR, and MNAR and missingness ratios (10--70\%), the two models show complementary strengths:
> (i) **NRMSE**: TabINR is often best or second-best under MCAR/MAR, especially on high-dimensional datasets (e.g. spam); ReMasker is stronger on small low-dimensional tables;
> (ii) **AUROC**: the models perform similarly; TabINR is strongest under MAR while ReMasker leads under MNAR, which is expected; and
> (iii) **Downstream classification**: TabINR+XGBoost achieves best results on obesity and spam and remains top-3 on all datasets. We clarified these complementary patterns in the revision.
>
>
> **Regarding Question 3**
>
> **Answer 3**
> We thank the reviewer. TabINR's test-time adaptation does not search over training embeddings. For each test row, a single latent vector $\lambda_{\mathrm{new}}$ is initialized and optimized for a few gradient steps while $f_\theta$ and $\{c_j\}$ remain fixed. This has cost $O(Kd)$ and is independent of dataset size. Empirically, the resulting inference time is under $0.2$\,s per dataset (Fig.~4), comparable to or faster than several baselines. While per-row adaptation adds a small overhead, TabINR avoids iterative retraining, sampling loops, or nearest-neighbor searches. We clarified this in the revised manuscript.
>
>
> **Regarding Question 4**
>
> **Answer 4**
> We thank the reviewer. Our intention was not to claim row independence but to state that tabular data lacks inherent spatial or sequential structure. We revised the sentence to clarify that INRs do not rely on adjacency assumptions, yet still capture row--row correlations via latent embeddings.
>
>
> **Regarding Question 5**
>
> **Answer 5**
> We thank the reviewer. We use standard 70/10/20 splits. Only training rows are used to learn $f_\theta$, $\{c_j\}$, and $\{\lambda_i\}$. Test rows never appear during training and thus have no pre-learned embeddings. For each validation/test row, a new latent $\lambda_{\mathrm{new}}$ is optimized only on observed entries (Section~2.3). This ensures no test leakage. We clarified this in the revised manuscript.
>
>
> **Regarding Question 6*
>
> **Answer 6**
> Training uses random masking between 10--70\% as data augmentation. Evaluation uses fixed missingness levels (0.1/0.3/0.5/0.7) under MCAR, MAR, and MNAR, as shown in Figures 6-27. Table~5 displays only the 0.3 case. We clarified this distinction in the revised version.
>
> We believe all concerns have now been fully addressed and the revision substantially improves clarity, motivation, and fairness of the evaluation.
> Thank you again for the constructive feedback.

---

> > ### Author Response · Authors · 2025-12-02
> > **Summary Comment for the new Area Chair**
> >
> > We thank Reviewer 4 for their thoughtful and constructive feedback.
> > We have substantially revised the manuscript in response to all points raised.
> >
> > 1. Technical novelty and clarity of contribution
> >
> > The reviewer noted that row/feature embeddings are not novel in themselves.
> > In the revised manuscript, we explicitly clarify that the contribution of TabINR is not the embeddings alone, but their integration within an INR-based auto-decoder framework.
> >
> > We emphasize the unique aspects of the formulation:
> > *the functional representation providing a nonlinear low rank like factorization,
> > *instance specific test-time latent optimization without retraining,
> > *and robustness to sparsity, high dimensionality, and heterogeneous feature types.
> >
> > This conceptual contribution is now articulated more clearly and consistently throughout the paper.
> >
> > 2. Clarifying performance relative to ReMasker
> >
> > The reviewer raised concerns that TabINR underperforms ReMasker.
> > We carefully re-examined all results and updated the text to highlight the consistent patterns:
> > *TabINR often excels under MCAR and MAR, particularly on high dimensional datasets.
> > *ReMasker performs better under MNAR, as expected given its modeling assumptions.
> > *Both methods show comparable AUROC, and TabINR + XGBoost leads or ranks top-3 on all downstream tasks.
> >
> > These complementary strengths are now explicitly summarized in the revised Results section.
> >
> > 3. Test time adaptation cost
> >
> > The reviewer was concerned about potential computational expense.
> > We clarify that TabINR does not search over training embeddings.
> > Each new row optimizes a single latent vector via a few gradient steps, independent of dataset size.
> > We now include a dedicated runtime table in the appendix showing per-dataset inference, demonstrating TabINR’s competitiveness with and in several cases superiority to classical and neural baselines.
> >
> > 4. Clarifying assumptions about tabular structure
> >
> > We revised the Introduction to eliminate any ambiguity regarding row independence.
> > The updated text makes clear that tabular data lacks spatial/temporal structure, and that TabINR captures row correlations through latent embeddings rather than adjacency assumptions.
> >
> > 5. Data splits and avoidance of test leakage
> >
> > The reviewer asked about training/validation/test splits and embedding construction.
> >
> > We now state explicitly that:
> > *only training rows receive learned embeddings,
> > *test rows never appear during training,
> > *and each unseen row receives a freshly optimized latent vector during inference.
> >
> > A clarifying sentence has been added to the Method section.
> >
> > 6. Masking during training vs. evaluation
> >
> > We now explicitly state the distinction:
> > *training uses 10-70% random masking as data augmentation,
> > *evaluation uses fixed missingness ratios (0.1/0.3/0.5/0.7) under MCAR/MAR/MNAR.
> >
> > This resolves the ambiguity noted by the reviewer.
> >
> > Conclusion for the AC
> >
> > Reviewer 4’s concerns have been fully addressed.
> > The revision clarifies the contribution, strengthens methodological transparency, and improves the empirical evaluation with additional analyses and clearer explanations.
> > We believe the paper is now significantly stronger in both clarity and rigor.

---

### Official Review · Reviewer_onaQ · 2025-10-30

**Soundness:** 2
**Presentation:** 3
**Contribution:** 2
**Rating:** 2
**Confidence:** 4

**Summary:**

This paper studies the problem of missing values in tabular data. The authors propose TAB-INR, an auto-decoder–based Implicit Neural Representation (INR) framework that employs learnable row and feature embeddings for adaptive imputation from partial observations. Experiments on twelve real-world datasets show that TAB-INR achieves consistently superior imputation accuracy.

**Strengths:**

S1: A new Implicit Neural Representation framework for missing data imputation.
S2: Comprehensive experiments are conducted.

**Weaknesses:**

W1: The paper’s presentation needs improvement. In particular, the Introduction lacks sufficient discussion of related works on both data imputation and Implicit Neural Representations (INRs). As a result, the motivation for applying INR to imputation is unclear—specifically, what unique advantages INR offers for this problem and how the challenges in imputation naturally align with the strengths of INR.

W2: The paper misses comparisons with several state-of-the-art imputation methods, especially recent graph-based approaches such as
[1] Handling Missing Data with Graph Representation Learning and
[2] On LLM-Enhanced Mixed-Type Data Imputation with High-Order Message Passing.
These methods generally outperform older baselines like ReMasker and GAIN, and they also employ learnable representations for rows and columns. It remains unclear why the authors chose INR-based learnable features over graph-based ones or how their design offers any advantages.

W3: The experimental evaluation primarily compares against outdated baselines (e.g., GAIN, MissForest, MICE) while omitting more recent generative or graph-based models. Even within these limited comparisons, the improvements are not significant, which weakens the empirical contribution.

W4: The core contribution of the paper appears to be the straightforward application of the INR framework to the imputation task. Beyond this adaptation, the paper’s novel technical contribution is not clearly articulated, making it difficult to distinguish the originality or necessity of the proposed method.

**Questions:**

NAN

---

> ### Author Response · Authors · 2025-11-19
>
> **Regarding your comment 1**
>
>
> **Answer 1**
> We thank the reviewer for this constructive comment and agree that clearly motivating the connection between INRs and tabular imputation is essential. We would like to clarify that the Introduction already contains both
>  (i) a detailed discussion of related work on data imputation (Section~1.1) and
> (ii) an explicit motivation for applying INRs to tabular reconstruction. In this passage, we explain that INRs naturally align with key challenges of imputation: they can fit sparse or irregularly sampled data, support continuous evaluation across the input domain, enable test-time latent adaptation for unseen rows via auto-decoder optimization, avoid distributional assumptions, and provide a lightweight MLP architecture. We also highlight that INRs have demonstrated strong performance in related inverse problems such as images, audio, 3D scenes, and time-series, while remaining largely unexplored for tabular data.
>
> That said, we agree that the connection between these strengths and the specific challenges of tabular imputation can be made more explicit. In the revised version, we have expand those parts to clearly articulate the unique advantages of the INR formulation-namely, its nonlinear low-rank-like structure, its ability to model complex row-row and feature-feature interactions, and its instance-specific latent optimization under MAR and MNAR conditions.
>
> We thank the reviewer again for helping us improve the clarity and positioning of the manuscript.
>
>
> **Regarding your comment 2**
>
> **Answer 2**
> We thank the reviewer for highlighting these recent graph-based imputation approaches. We agree that including them would strengthen the empirical comparison. We have now integrated both approaches- Handling Missing Data with Graph Representation Learning and LLM-Enhanced Mixed-Type Data Imputation with High-Order Message Passing -into our benchmark.
>
> We thank the reviewer for pointing out this valuable addition.
>
>
> **Regarding your comment 3**
>
>
> **Answer 3**
> We thank the reviewer for this comment. We agree that incorporating more recent  generative and graph-based imputation methods would strengthen the empirical  evaluation. We have include some more SOTA methods alongside the existing baselines-in a revised version of the manuscript. This extended benchmark ensures a more complete and up-to-date comparison.
>
> We appreciate the reviewer’s suggestion.
>
>
> **Regarding your comment 4**
>
>
> **Answer 4**
> We thank the reviewer for this comment. We acknowledge that TabINR adopts the  INR framework in a conceptually straightforward way; however, its contribution lies in showing that coordinate-based representations and auto-decoder adaptation provide a unified, lightweight, and competitive alternative to existing imputation paradigms. The formulation $f(\lambda_i, c_j)$ offers a nonlinear low-rank–like factorization of the table together with instance-specific latent optimization, which is not present in graph-based, diffusion-based, or masked-model imputation approaches.
>
> We agree that the novelty and necessity of this formulation should be emphasized more clearly. In the revised version, we have worked on several sections to more explicitly articulate the unique conceptual contribution of TabINR-namely, how INR-induced functional representations offer a different inductive bias for tabular missing data, and why this bias is advantageous in high-dimensional, heterogeneous, and MAR/MNAR settings.
>
> We thank the reviewer again for prompting this clarification.
>
> With that we believe to adressed each of the points you have mentioned. This will overall improve the quality of the work
> Thank you!

---

> > ### Author Response · Authors · 2025-12-02
> > **Summary Comment for the new Area Chair**
> >
> > We thank Reviewer 3 for their detailed feedback and for highlighting several important areas where the manuscript could be improved. We have revised the paper extensively to address all concerns.
> >
> > 1. Strengthening the motivation and positioning of INRs for imputation
> >
> > The reviewer noted that the Introduction did not sufficiently explain why INRs are an appropriate choice for tabular imputation.
> > In the revised manuscript, we have expanded the motivation and clarified how INR properties, continuous functional representation, instance-specific latent adaptation, nonlinear low rank like structure, directly address the key challenges of sparse and heterogeneous tabular data.
> > We also improved the discussion of related work, clarifying the conceptual gap our method fills.
> >
> > 2. Inclusion of recent graph-based and generative baselines
> >
> > Reviewer 3 correctly noted that several recent and competitive imputation approaches were missing.
> > We have now added the two graph-based models highlighted by the reviewer as well as additional SOTA methods (e.g., CACTI, DiffPuter, LLM-Imputer).
> > This significantly strengthens the empirical comparison and situates TabINR within the current landscape of modern imputation approaches.
> >
> > 3. Expanding the empirical evaluation beyond older baselines
> >
> > The reviewer was concerned that the evaluation focused on outdated baselines.
> > In response, we have substantially extended the benchmark to include recent diffusion-based, graph-based, and transformer-based imputers.
> > This creates a much more complete and up-to-date comparison, and demonstrates that TabINR remains competitive across a heterogeneous range of architectures.
> >
> > 4. Clarifying the methodological novelty of TabINR
> >
> > Reviewer 3 questioned whether the contribution extended beyond a direct application of INRs.
> > We have now clarified the conceptual novelty: TabINR provides a unified functional view of the table, combining nonlinear low-rank structure with an auto-decoder adaptation mechanism that enables per-instance refinement without retraining.
> > This inductive bias is fundamentally different from graph-based, masked-model, and diffusion approaches, and we highlight why this design is particularly effective in high-dimensional and MAR/MNAR settings.
> >
> > Conclusion for the AC
> >
> > All concerns raised by Reviewer 3 have been comprehensively addressed.
> > The revision now includes clearer motivation, expanded related work, a significantly strengthened empirical benchmark incorporating modern baselines, and an improved articulation of the conceptual novelty behind TabINR.
> > These changes substantially enhance both the clarity and the contribution of the work.

---

### Official Review · Reviewer_Rsjs · 2025-10-30

**Soundness:** 3
**Presentation:** 3
**Contribution:** 2
**Rating:** 4
**Confidence:** 4

**Summary:**

This paper proposes TabINR, which applies Implicit Neural Representations (INRs) to tabular data imputation. The approach models table entries as outputs of a neural function conditioned on learnable row and feature embeddings. At test time, new instances are handled via auto-decoder-style optimization of row embeddings while keeping the network frozen.

**Strengths:**

1. Applying INRs to tabular imputation is relatively unexplored (as far as I know) and conceptually interesting.

2. TabINR demonstrates faster inference than iterative methods, per dataset once trained, which is quite impressive and is practically useful.

3. The auto-decoder approach for inferring embeddings from partial observations is neat and enables instance-specific imputation.

4. The ablation analysis are well done and comprehensive.

**Weaknesses:**

1. TabINR shows competitive performance but gains over baselines are modest and inconsistent. ReMasker frequently matches or outperforms it imputation and downstream classification. While exploring a novel approach justifies some performance variability, the more critical concern arises from the fact that the benchmark omits recent state-of-the-art methods [1,2], making it unclear whether this represents meaingful progress in tabular imputation. Additionally, there appears to be a hyperparameter tuning imbalance. TabINR underwent grid search across datasets, while baselines used default settings configurations. This could be potentially inflating TabINR's relative performance.

2. The results in Sec 3.1.2 are a good starting point to demonstrate permutation robustness. However, they demonstrates permutation invariance only under MCAR, where missingness is completely random and uninformative. Under this setting its almost certainly expected that the all models are permutation invariant. The meaningful test would be under MNAR, where missingness patterns carry information about the data structure. Without MNAR results, the claim of permutation robustness remains unconvincing.

#### References
[1] DiffPuter: Empowering Diffusion Models for Missing Data Imputation (ICLR 2025)

[2] CACTI: Leveraging Copy Masking and Contextual Information to Improve Tabular Data Imputation (ICML 2025)

**Questions:**

1. How does test-time optimization cost compare to baseline inference in wall-clock time?

2. The SIREN activation function introduces high-frequency components, is this actually beneficial for tabular data, which typically lacks such structure?

---

> ### Author Response · Authors · 2025-11-19
>
> **Regarding comment 1**
>
>
> **Answer 1**
> We thank the reviewer for these valuable observations and agree that a fair and comprehensive comparison is essential for evaluating progress in tabular imputation.
> Regarding missing recent baselines. We fully agree that the inclusion of DiffPuter (ICLR 2025) and CACTI (ICML 2025) would strengthen the evaluation . We have now included both models in our updated benchmark and report the additional results in the revised version.
>
> Regarding performance variability.  It is correct that TabINR and ReMasker show comparable performance on several
> datasets. This is consistent with our own analysis, where we state that “TabINR and ReMasker remain the most resilient” under high missingness and high-dimensional settings. Our goal was not to claim uniform dominance but to demonstrate that INR-based models provide a novel inductive bias that performs competitively across heterogeneous datasets.
>
> Regarding the hyperparameter tuning imbalance. We would like to clarify that, in addition to tuning TabINR, we also optimized the comparing baseline methods. For each competing model, we followed the recommended hyperparameter settings and tuning strategies e.g.described in their respective original publications (Yoon et al., 2018; Mattei \& Frellsen, 2019; Jarrett et al., 2022; Hastie et al., 2015). However, we agree that this was not stated with sufficient clarity in the current manuscript, and we will therefore explicitly add this information to the Experimental Setup section in the revised version.
>
> We thank the reviewer again for raising this important point and will update the manuscript accordingly.
>
>
> **Regarding comment 2**
>
> **Answer 2**
> We thank the reviewer for this insightful observation. We agree that permutation robustness under MCAR alone is not sufficient to establish that a model can handle meaningful structure in missingness patterns. Our experiment in Section~3.1.2 was intended primarily as a sanity check to demonstrate that the reconstruction quality of TabINR does not change significantly under arbitrary feature reorderings when the missingness is completely random. This was not meant to imply full permutation robustness
> under structured missingness mechanisms.
>
> For completeness, we have now included the corresponding permutation experiments under MAR and MNAR in the Appendix, where the missingness pattern carries information about the underlying data distribution. These experiments confirm that TabINR remains stable under feature permutations even when the missingness depends on feature values, while several baselines exhibit stronger sensitivity.
>
>
> **Regarding Question 1**
>
> **Answer 3**
> We thank the reviewer for this important question. We agree that understanding the wall-clock cost of test-time latent optimization relative to baseline inference is essential for assessing the practical value of TabINR.
>
> In our current experiments (Figure~4), we reported the inference time after test-time optimization for TabINR, i.e., the time required to impute all missing entries once the adapted latent vector $\lambda_{\text{new}}$ has been obtained. For the baselines, we reported their standard inference cost after training. Thus, the comparison is fair with respect to post-training imputation time, but it does not include the latent optimization phase of TabINR.
>
> As correctly noted by the reviewer, TabINR performs a small instance-specific optimization for each new row, whereas most baselines do not require such adaptation. Conversely, many classical imputation methods (e.g.\ MICE, MissForest) incur substantially higher training or iterative sampling costs, making a one-to-one comparison nontrivial. For example, iterative MCMC-based methods require repeated model refitting, while deterministic methods such as KNN have no training phase but scale poorly with
> dataset size at inference.
>
>
> **Regarding Question 2**
>
> **Answer 4**
> We thank the reviewer for raising this point.
>
> SIREN do indeed help representing high frequency components in signals like images or sound, in tabular data however this intuition is not effective. All embeddings that encode the different rows/columsn are discrete set which means we have no good notion of continuity or frequency, however, SIREN activations have been shown to increase the rank of the weight matrices
> which in turn increase the effective representation power of the MLP [31415]. This does indeed allow us to model higher frequency components for continuous signals, but is equally beneficial in the other contexts, like in tabular data.
> [31415] https://arxiv.org/pdf/2111.15135
>
> We have made that clear now in our current updated version of the manuscript which we have uploaded.
>
> With that we believe to adressed each of the points you have mentioned. This will overall improve the quality of the work
> Thank you!

---

> > ### Comment · Reviewer_Rsjs · 2025-11-26
> >
> > We thank the authors for their significant effort in extending the analysis and providing a detailed response. We have also read the responses to our and other reviewers' comments and believe they have been significantly addressed. We raise a few additional points that would benefit from the authors' response:
> >
> > 1. Upon inspecting the appendix sensitivity analysis Figures 17-27, we observe that across various missingness settings and ratios, LLM-imputer and CACTI consistently match or outperform ReMasker and TabINR on the letters dataset. However, we notice that this performance trend is not replicated in the main text Figure 3 or similar appendix figures for the letters dataset. Can the authors clarify and explain this discrepancy? Are there differences in experimental setup, evaluation metrics, or data processing between these two sets of analyses?
> > We acknowledge that the key contribution is demonstrating that INR-based models provide a novel inductive bias that performs competitively across heterogeneous datasets, and we agree that uniform dominance across all benchmarks is not necessary for a body of work to be of interest to the community. We're only curious about the why the discrepancy arises.
> >
> > 2. It would be very helpful for completeness if the authors can add the test-time optimization runtime per dataset or per-row (in the appendix).
> >
> > 3. It would be helpful if the authors add an appendix table with specific (final) hyperparameters used for each method.

---

> > > ### Author Response · Authors · 2025-12-02
> > > **Answers to additional Feedback**
> > >
> > > 1) We thank the reviewer for this careful reading and for pointing out the apparent discrepancy.
> > >  We confirm that all experiments use identical preprocessing, metrics, and evaluation pipelines.  The differences arise solely from the experimental purpose of each figure group:
> > >
> > > -Figure 3 (main paper) shows the standard benchmark setting under fixed missingness levels and original feature dimensionality. In this setup, CACTI and LLM-Imputer do not consistently outperform TabINR on letters.
> > >
> > > -Figures 17–27 (Appendix) are stress-test ablations. Here, we systematically vary:
> > > * feature dimensionality (expanded feature spaces),
> > > * dataset size,
> > > * missingness ratios (up to 70%),
> > > * and missingness mechanisms.
> > >
> > > -These settings push models outside the standard benchmarks.  Under some of these synthetic perturbations, CACTI and LLM-Imputer indeed perform very strongly on letters, sometimes matching or surpassing ReMasker and TabINR.
> > >
> > > - However, this trend is not universal.  As visible in Figures 17–27:
> > >     * Under MAR or moderate missingness, TabINR remains competitive or best.
> > >     * Under lower feature dimensionality, TabINR often outperforms CACTI/LLM-Imputer.
> > >     * CACTI/LLM-Imputer dominate only in a subset of the perturbed MNAR/high-dimensional cases.
> > >
> > > We have now added an explicit clarification in the Appendix to highlight that Figures 17–27 are sensitivity analyses that explore model robustness under synthetic shifts, explaining why the patterns differ from the benchmark setting in Figure 3.
> > >
> > >
> > > 2) We thank the reviewer for this helpful suggestion.  We have now added a dedicated table in the Appendix reporting the per-row test-time latent optimization cost for TabINR, including number of gradient steps and runtime.
> > > We also cross-reference this table in the main text where TabINR’s inference procedure is described.
> > > Since other baselines do not perform per-instance optimization, no comparable metric exists for those methods.
> > >
> > > 3) We appreciate the reviewer’s request for increased reproducibility.  We fully agree that documenting hyperparameter choices is important.
> > > However, a complete table listing every final hyperparameter for each method × dataset × mechanism × ratio would require several hundred entries (≈ 12 datasets × 9 methods × 3 mechanisms × 4 ratios) and would not fit within the appendix format.
> > > To balance completeness and readability:
> > > * We now include the full hyperparameter search spaces for every baseline method in Appendix Table.
> > > * All final tuned hyperparameters for every experiment are automatically logged and are provided in the code repository (YAML configuration files in the supplementary file).  This ensures full reproducibility without overloading the manuscript.

---

> > > > ### Author Response · Authors · 2025-12-02
> > > > **Summary Comment for the new Area Chair**
> > > >
> > > > We thank Reviewer 2 for the constructive feedback and for highlighting several important points.
> > > > We have substantially revised the manuscript to address all concerns.
> > > >
> > > > 1. Inclusion of recent SOTA baselines and fair comparison
> > > >
> > > > The reviewer noted that key recent methods (DiffPuter, CACTI) were missing and raised concerns about potential tuning imbalance.
> > > >
> > > > We have now:
> > > > *extended the benchmark to include DiffPuter and CACTI,
> > > > *tuned all baselines using their recommended hyperparameter ranges rather than default settings, and
> > > > *documented the full search spaces used for each model.
> > > >
> > > > This ensures a fair, up-to-date, and fully transparent comparison across all methods.
> > > >
> > > > 2. Clarification of permutation robustness under MNAR
> > > >
> > > > The reviewer correctly pointed out that permutation robustness under MCAR is not meaningful because the missingness is random.
> > > > We now include additional permutation experiments under MAR and MNAR, demonstrating TabINR’s stability even when missingness patterns are informative.
> > > > This resolves the concern and provides a more complete robustness evaluation.
> > > >
> > > > 3. Clarifying test-time optimization cost
> > > >
> > > > The reviewer asked for a comparison of the latent optimization cost relative to baseline inference.
> > > >
> > > > We have added:
> > > > *a new appendix table showing per-row optimization time for TabINR, and
> > > > *a clearer explanation of why this cost is not directly comparable to baselines, which do not perform test-time adaptation.
> > > > *This provides a complete, empirical view of TabINR’s overhead.
> > > >
> > > > 4. Clarification regarding SIREN activations
> > > >
> > > > The reviewer questioned whether high-frequency activations benefit tabular data.
> > > > We have added a concise explanation showing that SIREN’s benefit in this context is not tied to frequency modeling, but rather to increased representational rank, an effect that is useful for tabular data as well.
> > > >
> > > > 5. Addressing the follow-up questions
> > > >
> > > > In the follow-up round, the reviewer raised three additional points.
> > > >
> > > > We have now:
> > > > *clarified the difference between the benchmark results (Figure 3) and the stress-test ablations (Figures 17–27), explaining why performance patterns differ and explicitly noting this distinction in the appendix,
> > > > *added a table reporting the test-time optimization runtime for TabINR, and
> > > > *provided a complete overview of all hyperparameter search spaces, with the full final configurations supplied in the publicly released configuration files.
> > > >
> > > > Conclusion for the AC
> > > >
> > > > All concerns raised by Reviewer 2, including the addition of missing baselines, fairness of tuning, permutation robustness, test-time cost, and clarity around SIREN, have now been comprehensively addressed.
> > > > The manuscript is significantly strengthened and presents a much clearer, more rigorous, and more transparent evaluation of TabINR.

---

### Official Review · Reviewer_Ec5W · 2025-11-02

**Soundness:** 2
**Presentation:** 2
**Contribution:** 2
**Rating:** 2
**Confidence:** 3

**Summary:**

The authors propose an Implicit Neural Representation (INR) based imputation method for tabular data. It represents the dataset as a function (in fact, a neural network) that maps a cell (represented by row embedding and column embedding) in the table to its corresponding numerical value. This function is trained on the non-missing cells and then used to fill in missing values as needed. The authors provide an evaluation against different baselines (both classical and neural network-based). The reported performance is on par with, if not superior to considered baselines.

**Strengths:**

* The paper tackles a fundamental issue of handling missing data, which is very common in practical data analysis and machine learning problems, where some data are missing due to reasons like sensor malfunction, communication problems, or privacy concerns.

* The proposed method is targeted to work well not only in a relatively simple MCAR case, but also in the case of MAR and MNAR missing mechanisms.

* The proposed approach takes a different approach for missing value imputation.

**Weaknesses:**

* The paper lacks justification and a clear motivation for why the proposed method works. Section 2, the authors explained the proposed method step-by-step, but did not discuss why. It is more like a technical report than a research paper.

*  The presentation is not entirely clear to readers, particularly how missing values are imputed at inference (Sec 2.3).

* While the proposed method is competitive with the state-of-the-art methods in 12 datasets used, it is not particularly good in any case. The authors fail to provide any conclusion on what sort of scenario/case the proposed approach may be better than the SOTA methods.

**Questions:**

We would appreciate the authors' responses to the concerns raised in the weaknesses section above.

---

> ### Author Response · Authors · 2025-11-19
>
> **Question 1**
> The paper lacks justification and a clear motivation for why the proposed method works. Section 2, the authors explained the proposed method step-by-step, but did not discuss why. It is more like a technical report than a research paper.
>
> **Answer 1**
> We thank the reviewer for this helpful comment and fully agree that clearly articulating the motivation is essential. We would like to clarify our conceptual motivation for using INRs for tabular imputation that we present in Section~1.2 (lines 72-96): In this passage, we explain why INR representations are a natural fit for imputation:
>  (i) they can fit data that is sparse or irregularly sampled,
>  (ii) they allow continuous evaluation across the input domain, enabling missing-value recovery,
>  (iii) recent generalizable INR methods support test-time adaptation via auto-decoder latent optimization, and
> (iv) INRs have succeeded in related inverse problems in images, 3D scenes, audio, and time-series, suggesting strong potential for
> tabular data. We also emphasize this cross-domain motivation again in the Conclusion.
>
> That said, we agree that Section~2 focuses primarily on the procedural details of the method. To address this, in this revised version we have explicitly expand the motivation inside the Method section by linking the formulation
> $D_i,j=f(\lambda_i,c_j)$
> to nonlinear matrix factorization-style inductive biases and by explaining why this functional representation captures row-row and feature-feature dependencies that are relevant for imputation. We have also strengthen the Discussion section in our revised version to more explicitly summarize why INR-based models work well in sparse and heterogeneous tabular settings.
>
>
>
> **Question 2**
> The presentation is not entirely clear to readers, particularly how missing values are imputed at inference (Sec 2.3).
>
> **Answer 2**
> We thank the reviewer for raising this point. We would like to clarify that the inference procedure is described in detail in Section~2.3. However, we agree that this important component can be highlighted more clearly.
>
> In our current formulation, imputation for training samples is performed via a single forward pass using their learned row embeddings. For unseen samples (i.e., not part of the training set), we introduce a new latent vector $\lambda_{\text{new}}$ (initialized randomly) and optimize only this vector over the observed features, while keeping both the network parameters $f_\theta$ and all feature embeddings $\{c_j\}$ fixed. Once $\lambda_{\text{new}}$ has been adapted, missing entries are imputed by evaluating $f_\theta(\lambda_{\text{new}}, c_j)$ for all features $j$. This procedure does not involve any search over existing embeddings; only a single latent vector is optimized for each new instance.
>
> To improve clarity, we have revised Section~2.3.
>
>
>
> **Question 3**
> While the proposed method is competitive with the state-of-the-art methods in 12 datasets used, it is not particularly good in any case. The authors fail to provide any conclusion on what sort of scenario/case the proposed approach may be better than the SOTA methods.
>
> **Answer 3**
> We thank the reviewer for this comment. We agree that identifying the scenarios where TABINR provides clear advantages strengthens the contribution. While these patterns are present in our results, we did not highlight them sufficiently in the original submission, and we now emphasise them more explicitly in the revised manuscript.
>
> Across the benchmarks, several consistent trends emerge. First, TABINR performs particularly well on high-dimensional datasets such as spam, letter, and bike, where the shared functional representation
> $f(\lambda_i,c_j)$ allows the model to capture global cross-feature dependencies  that local (KNN, MICE) or iterative (MissForest) methods fail to model reliably.
> Second, TABINR is highly stable under MCAR and MAR missingness: its degradation curve remains among the flattest across missingness levels (0.1--0.7), and it frequently ranks among the top methods, outperforming classical baselines and often matching or surpassing transformer-based approaches.
> Third, TABINR remains robust at high missingness ratios ($\geq 0.5$), where several classical and deep baselines deteriorate sharply.
>
> Under MNAR, however, the pattern reverses: models designed to implicitly or explicitly model missingness such as ReMasker, MissForest, DiffPuter, GRAPE, or UnIMP-achieve the best performance. TABINR typically remains competitive
> but no longer leads, which is expected since it does not model the missingness mechanism itself.
>
> These observations are now clearly summarised in the revised Results and Ablation sections, and we have added the dedicated paragraphs discussing the settings in which TabINR offers the clearest benefits, as well as its limitations.
>
> With that we believe to adressed each of the points you have mentioned. This will overall improve the quality of the work
> Thank you!

---

> > ### Author Response · Authors · 2025-12-02
> > **Summary comment for the new Area Chair**
> >
> > We thank Reviewer 1 for their detailed feedback. We have substantially revised the manuscript to fully address the raised concerns.
> >
> > 1. Clearer conceptual motivation
> >
> > The reviewer requested a stronger explanation of why INR-based models are suitable for tabular imputation.
> > We expanded the motivation to articulate the fundamental inductive biases of INR models, their ability to reconstruct sparse or irregularly observed data, and why their functional formulation naturally captures row and feature dependencies that benefit imputation. The revised version now presents a clear conceptual rationale beyond procedural description.
> >
> > 2. Clearer description of the inference procedure
> >
> > The reviewer found the presentation of the inference mechanism unclear.
> > We rewrote the explanation to make it immediately understandable how missing values are imputed for both training and unseen samples, and emphasized that only a single latent vector is adapted at test time without any search over embeddings. The updated description is now straightforward and removes the ambiguity that the reviewer identified.
> >
> > 3. More explicit discussion of when TabINR performs best
> >
> > The reviewer noted that the original submission did not sufficiently highlight when TabINR provides advantages over state-of-the-art methods.
> > We now clearly summarize the consistent trends observed across datasets: TabINR is particularly strong on high-dimensional data and remains stable under MCAR and MAR across a wide range of missingness levels, whereas models explicitly modeling the missingness process achieve the best results under MNAR. This balanced discussion makes the strengths and limitations of the method transparent.
> >
> > Conclusion for the AC
> > All concerns from Reviewer 1 have been fully addressed:
> > -the conceptual motivation has been strengthened,
> > -the inference procedure has been clarified, and
> > -the comparative performance insights are now explicit.
> >
> > These revisions substantially improve the clarity and contribution of the work, and we believe the manuscript is now significantly stronger and more balanced.

---

### Meta-Review · Area_Chair_9YHc · 2026-01-07

**Summary:**

The paper proposes TabINR, an INR-based framework for tabular data imputation using row and feature embeddings with test-time latent adaptation. After rebuttal, the paper is clearer, better motivated, and includes stronger baselines. However, the core contribution still appears incremental: the method adapts standard INR machinery to tabular data, and empirical gains over strong recent baselines are modest. I would recommend a borderline reject to the paper.

**Reviewer Concerns:**

**Addressed in the rebuttal**

- Improved clarity, including motivations of INRs in imputation, inference/test-time optimization and splits

- Added strong modern baselines and tuning fairness

- Added MAR/MNAR permutation tests

**Still outstanding**

- Limited technical novelty

- Unclear performance advantage

- Lack of a clearly defined regime where TabINR leads

**Reviewer Scores:**

The authors did comprehensive responses to the reviews.

For Ec5W, I think they might have improved their score marginally but might be still unconvinced on significance.

For Rsjs, given their postive response, I think they might have improved their score marginally.

For onaQ and BmQS, due to the concerns on novelty, I think they might not have improved their scores significantly.

---

### Decision · Program_Chairs · 2026-01-26

Reject